# Transfer Learning with Deep Tabular Models

**Roman Levin**[*1‡]     **Valeriia Cherepanova**[*2]     **Avi Schwarzschild**[†2]     **Arpit Bansal**[†2]
**C. Bayan Bruss**[3]     **Tom Goldstein**[2]     **Andrew Gordon Wilson**[4]     **Micah Goldblum**[4]
[1]University of Washington   [2]University of Maryland   [3]Capital One   [4]New York University
rilevin@uw.edu, vcherepa@umd.edu

## Abstract

Recent work on deep learning for tabular data demonstrates the strong performance of deep tabular models, often bridging the gap between gradient boosted decision trees and neural networks. Accuracy aside, a major advantage of neural models is that they are easily fine-tuned in new domains and learn reusable features. This property is often exploited in computer vision and natural language applications, where transfer learning is indispensable when task-specific training data is scarce. In this work, we explore the benefits that representation learning provides for knowledge transfer in the tabular domain. We conduct experiments in a realistic medical diagnosis test bed with limited amounts of downstream data and find that transfer learning with deep tabular models provides a definitive advantage over gradient boosted decision tree methods. We further compare the supervised and self-supervised pre-training strategies and provide practical advice on transfer learning with tabular models. Finally, we propose a pseudo-feature method for cases where the upstream and downstream feature sets differ, a tabular-specific problem widespread in real-world applications.

## 1 Introduction

Tabular data is ubiquitous throughout diverse real-world applications, spanning medical diagnosis (Johnson et al., 2016), housing price prediction (Afonso et al., 2019), loan approval (Arun et al., 2016), and robotics (Wienke et al., 2018), yet practitioners still rely heavily on classical machine learning systems. Recently, neural network architectures and training routines for tabular data have advanced significantly. Leading methods in tabular deep learning (Gorishniy et al., 2021; 2022; Somepalli et al., 2021; Kossen et al., 2021) now perform on par with the traditionally dominant gradient boosted decision trees (GBDT) (Friedman, 2001; Prokhorenkova et al., 2018; Chen and Guestrin, 2016; Ke et al., 2017). On top of their competitive performance, neural networks, which are end-to-end differentiable and extract complex data representations, possess numerous capabilities which decision trees lack; one especially useful capability is *transfer learning*, in which a representation learned on *pre-training* data is reused or fine-tuned on one or more *downstream* tasks.

Transfer learning plays a central role in industrial computer vision and natural language processing pipelines, where models learn generic features that are useful across many tasks. For example, feature extractors pre-trained on the ImageNet dataset can enhance object detectors (Ren et al., 2015), and large transformer models trained on vast text corpora develop conceptual understandings which can be readily fine-tuned for question answering or language inference (Devlin et al., 2019). One might wonder if deep neural networks for tabular data, which are typically shallow and whose hierarchical feature extraction is unexplored, can also build representations that are transferable beyond their pre-training tasks. In fact, a recent survey paper on deep learning with tabular data suggested that efficient knowledge transfer in tabular data is an open research question (Borisov et al., 2021). In this work, we show that deep tabular models with transfer learning definitively outperform their classical counterparts when auxiliary upstream pre-training data is available and the amount of downstream data is limited. Importantly, we find representation learning with tabular neural networks to be more powerful than gradient boosted decision trees with stacking – a strong baseline leveraging knowledge transfer from the upstream data with classical methods.

---

[*†] Equal contribution. [‡] This work was completed prior to the author joining Amazon

Some of the most common real-world scenarios with limited data are medical applications. Accumulating large amounts of patient data with labels is often very difficult, especially for rare conditions or hospital-specific tasks. However, large related datasets, e.g. for more common diagnoses, may be available in such cases. We note that while computer vision medical applications are common (Irvin et al., 2019; Santa Cruz et al., 2021; Chen et al., 2018b; Turbé et al., 2021), many medical datasets are fundamentally tabular (Goldberger et al., 2000; Johnson et al., 2021; 2016; Law and Liu, 2009). Motivated by this scenario, we choose a realistic medical diagnosis test bed for our experiments both for its practical value and transfer learning suitability. We first design a suite of benchmark transfer learning tasks using the MetaMIMIC repository (Grzyb et al., 2021; Woźnica et al., 2022) and use this collection of tasks to compare transfer learning with prominent tabular models and GBDT methods at different levels of downstream data availability. We explore several transfer learning setups and lend suggestions to practitioners who may adopt tabular transfer learning. Additionally, we compare supervised pre-training and self-supervised pre-training strategies and find that supervised pre-training leads to more transferable features in the tabular domain, contrary to findings in vision where a mature progression of self-supervised methods exhibit strong performance (He et al., 2020).

Finally, we propose a pseudo-feature method which enables transfer learning when upstream and downstream feature sets differ. As tabular data is highly heterogeneous, the problem of downstream tasks whose formats and features differ from those of upstream data is common and has been reported to complicate knowledge transfer (Lewinson, 2020). Nonetheless, if our upstream data is missing columns present in downstream data, we still want to leverage pre-training. Our approach uses transfer learning in stages. In the case that upstream data is missing a column, we first pre-train a model on the upstream data without that feature. We then fine-tune the pre-trained model on downstream data to predict values in the column absent from the upstream data. Finally, after assigning pseudo-values of this feature to the upstream samples, we re-do the pre-training and transfer the feature extractor to the downstream task. This approach offers appreciable performance boosts over discarding the missing features and often performs comparably to models pre-trained with the ground truth feature values.

Our contributions are summarized as follows:

1. We find that recent deep tabular models combined with transfer learning have a decisive advantage over strong GBDT baselines, even those that also leverage upstream data.
2. We compare supervised and self-supervised pre-training strategies and find that the supervised pre-training leads to more transferable features in the tabular domain.
3. We propose a pseudo-feature method for aligning the upstream and downstream feature sets in heterogeneous data, addressing a common obstacle in the tabular domain.
4. We provide advice for practitioners on architectures, hyperparameter tuning, and transfer learning setups for tabular transfer learning.

## 2 RELATED WORK

**Deep learning for tabular data.** The field of machine learning for tabular data has traditionally been dominated by gradient-boosted decision trees (Friedman, 2001; Chen and Guestrin, 2016; Ke et al., 2017; Prokhorenkova et al., 2018). These models are used for practical applications across domains ranging from finance to medicine and are consistently recommended as the approach of choice for modeling tabular data (Shwartz-Ziv and Armon, 2022).

An extensive line of work on tabular deep learning aims to challenge the dominance of GBDT models. Numerous tabular neural architectures have been introduced, based on the ideas of creating differentiable learner ensembles (Popov et al., 2019; Hazimeh et al., 2020; Yang et al., 2018; Kontschieder et al., 2015; Badirli et al., 2020), incorporating attention mechanisms and transformer architectures (Somepalli et al., 2021; Gorishniy et al., 2021; Arık and Pfister, 2021; Huang et al., 2020; Song et al., 2019; Kossen et al., 2021), as well as a variety of other approaches (Wang et al., 2017; 2021; Beutel et al., 2018; Klambauer et al., 2017; Fiedler, 2021; Schäfl et al., 2021). However, recent systematic benchmarking of deep tabular models (Gorishniy et al., 2021; Shwartz-Ziv and Armon, 2022) shows that while these models are competitive with GBDT on some tasks, there is still no universal best method. Gorishniy et al. (2021) show that transformer-based models are the strongest

alternative to GBDT and that ResNet and MLP models coupled with a strong hyperparameter tuning routine (Akiba et al., 2019) offer competitive baselines. Similarly, Kadra et al. (2021) and Fiedler (2021) find that carefully regularized and slightly modified MLPs are competitive as well as other regularized architectures (Yang et al., 2022). In a follow-up work, Gorishniy et al. (2022) show that transformer architectures equipped with advanced embedding schemes for numerical features bridge the performance gap between deep tabular models and GBDT.

**Transfer learning.** Transfer learning (Pan and Yang, 2010; Weiss et al., 2016; Zhuang et al., 2020) has been incredibly successful in domains of computer vision and natural language processing (NLP). Large fine-tuned models excel on a variety of image classification (Dosovitskiy et al., 2020; Dai et al., 2021) and NLP benchmarks (Devlin et al., 2019; Howard and Ruder, 2018). ImageNet (Deng et al., 2009) pre-trained feature extractors are incorporated into the complex pipelines of successful object detection and semantic segmentation models (Chen et al., 2018a; Ren et al., 2015; Redmon and Farhadi, 2018; Redmon et al., 2016). Transfer learning is also particularly helpful in applications with limited data availability such as medical image classification (Alzubaidi et al., 2021; Heker and Greenspan, 2020; Chen et al., 2019; Alzubaidi et al., 2020).

In the tabular data domain, a recent review paper (Borisov et al., 2021) finds that transfer learning is underexplored and that the question of how to perform knowledge transfer and leverage upstream data remains open. In our work, we seek to answer these questions through a systematic study of transfer learning with recent successful deep tabular models.

Multiple works mention that transfer learning in the tabular domain is challenging due to the highly heterogeneous nature of tabular data (Jain et al., 2021; Lewinson, 2020). Several papers focus on converting tabular data to images instead (Sharma et al., 2019; Zhu et al., 2021; Sun et al., 2019) and leveraging transfer learning with vision models (Sun et al., 2019). Other studies explore designing CNN-like inductive biases for tabular models (Joffe, 2021), transferring XGBoost hyperparameters (Woźnica et al., 2022; Grzyb et al., 2021), and transferring whole models (Fang et al., 2019; Al-Stouhi and Reddy, 2011; Li et al., 2021) in the limited setting of shared label and feature space between the upstream and downstream tasks. Additionally, a recent work by Wang and Sun (2022) proposes a variable-column tabular neural network architecture and applies it to learn information from multiple tables with overlapping columns. While TransTab operates on tables with shared label space, our work focuses on exploring how representation learning can be used to transfer knowledge between related, but different tasks.

Stacking could also be seen as a form of leveraging upstream knowledge in classical methods (Wolpert, 1992; Ting and Witten, 1997).

**Self-supervised learning.** Self-supervised learning (SSL) aimed at harnessing unlabelled data through learning its structure and invariances has accumulated a large body of works over the last few years. Prominent SSL methods, such as Masked Language Modeling (MLM) (Devlin et al., 2019) in NLP and contrastive pre-training in computer vision (Chen et al., 2020) have revolutionized their fields making SSL the pre-training approach of choice (Devlin et al., 2019; Lan et al., 2019; Liu et al., 2019; Lewis et al., 2019; Chen et al., 2020; He et al., 2020; Caron et al., 2020; Bardes et al., 2021; Misra and Maaten, 2020). In fact, SSL pre-training in vision has been shown to produce more transferable features than supervised pre-training on ImageNet (He et al., 2020).

Recently, SSL has been adopted in the tabular domain for semi-supervised learning (Yin et al., 2020; Yoon et al., 2020; Ucar et al., 2021; Somepalli et al., 2021; Huang et al., 2020). Contrastive pre-training on auxilary unlabelled data (Somepalli et al., 2021) and MLM-like approaches (Huang et al., 2020) have been shown to provide gains over training from scratch for transformer tabular architectures in cases of limited labelled data. Additionaly, Rubachev et al. (2022) investigate benefits of supervised and unsupervised pre-training when applied to the same data without transferring knowledge between tasks. Finally, self-supervised pre-training was successfully applied to time-series data in medical domain (Yèche et al., 2021; McDermott et al., 2021).

## 3   Transfer Learning Setup in Tabular Domain

To study transfer learning in the tabular domain, we need to choose benchmark tasks and training pipelines. In this section, we detail both our upstream-downstream datasets as well as the tools we use to optimize transfer learning for tabular data.

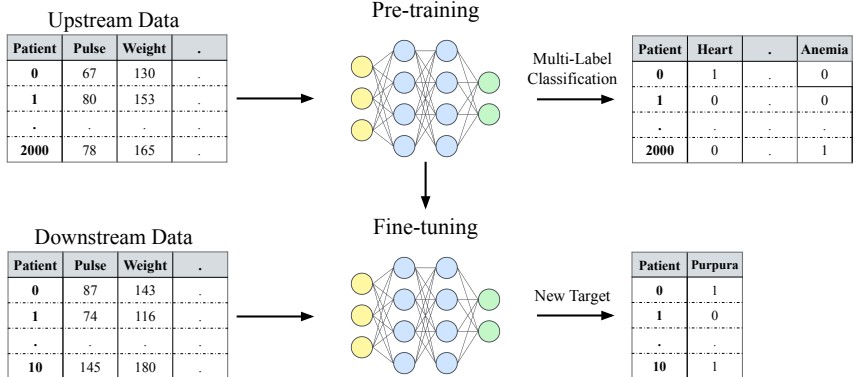

Figure 1: **Tabular transfer learning pipeline with MetaMIMIC.** We pre-train deep tabular neural networks on abundant upstream patient data with 11 diagnosis targets via multi-label classification. Then, we fine-tune the pre-trained models on limited downstream data with similar features to predict the new target diagnosis.

### 3.1 METAMIMIC TEST BED FOR TRANSFER LEARNING EXPERIMENTS

**MetaMIMIC repository.** As medical diagnosis data often contains similar test results (i.e. features) across patients, and some diseases (i.e. tasks) are common while others are rare, this setting is a realistic use-case for our work. We thus construct a suite of transfer learning benchmarks using the MetaMIMIC repository Grzyb et al. (2021); Woźnica et al. (2022) which is based on the MIMIC-IV Johnson et al. (2021); Goldberger et al. (2000) clinical database of anonymized patient data from the the Beth Israel Deaconess Medical Center ICU admissions. MetaMIMIC contains 12 binary prediction tasks corresponding to different diagnoses such as hypertensive diseases, ischematic heart disease, diabetes, alcohol dependence and others. It covers 34925 patients and 172 features, including one categorical feature (gender), related to the medical examination of patients hand-selected by the authors to have the smallest number of missing values (Grzyb et al., 2021; Woźnica et al., 2022). The features include mean, maximum and minimum statistics of lab test results as well as general features such as height, weight, age and gender. The 12 medical diagnosis targets are related tasks of varied similarity and make MetaMIMIC a perfect test bed for transfer learning experiments.

**Upstream and downstream tasks.** A medical practitioner may possess abundant annotated diagnosis data corresponding to a number of common diseases and want to harness this data to assist in diagnosing another disease, perhaps one which is rare or for which data is scarce. To simulate this scenario, we create transfer learning problems by splitting the MetaMIMIC data into upstream and downstream tasks. Specifically, we reserve 11 targets for the upstream pre-training tasks and one target for the downstream fine-tuning tasks, thus obtaining 12 upstream-downstream splits – one for each downstream diagnosis. Additionally, we limit the amount of downstream data to 4, 10, 20, 100, and 200 samples corresponding to several scenarios of data availability.

In total, we have 60 combinations of upstream and downstream datasets for our transfer learning experiments. We pre-train our models as multi-label classifiers on the upstream datasets with 11 targets and then transfer the feature extractors onto the downstream binary diagnosis tasks, Figure 1 presents a diagram illustrating the pipeline.

### 3.2 TABULAR MODELS

We conduct transfer learning experiments with four tabular neural networks, and we compare them to two GBDT implementations.

For neural networks, we use transformer-based architectures found to be the most competitive with GBDT tabular approaches (Gorishniy et al., 2021; Huang et al., 2020; Gorishniy et al., 2022). The specific implementations we consider include the recent FT-Transformer (Gorishniy et al., 2021) and TabTransformer (Huang et al., 2020). We do not include implementations with inter-sample attention

(Somepalli et al., 2021; Kossen et al., 2021) in our experiments since these do not lend themselves to scenarios with extremely limited downstream data. In addition, we use MLP and ResNet tabular architectures as they are known to be consistent and competitive baselines (Gorishniy et al., 2021).

The TabTransformer architecture comprises of an embedding layer for categorical features, a stack of transformer layers and a multi-layer perceptron applied to concatenation of processed categorical features and normalized numerical features. In contrast, FT-Transformer architecture transforms all features (including numerical) to embeddings and applies a stack of Transformer layers to the embeddings.

For GBDT implementation, we use the popular Catboost (Prokhorenkova et al., 2018) and XGBoost libraries (Chen and Guestrin, 2016).

### 3.3 Transfer Learning Setups and Baselines

In addition to a range of architectures, we consider several setups for transferring the upstream pre-trained neural feature extractors onto downstream tasks. Specifically, we use either a single linear layer or a two-layer MLP with 200 neurons in each layer for the classification head. We also either freeze the weights of the feature extractor or fine-tune the entire model end-to-end. To summarize, we implement four transfer learning setups for neural networks:

- Linear head atop a frozen feature extractor
- MLP head atop a frozen feature extractor
- End-to-end fine-tuned feature extractor with a linear head
- End-to-end fine-tuned feature extractor with an MLP head

We compare the above setups to the following baselines:

- Neural models trained from scratch on downstream data
- CatBoost and XGBoost with and without stacking

We use stacking for GBDT models to build a stronger baseline which leverages the upstream data (Wolpert, 1992; Ting and Witten, 1997). To implement stacking, we first train upstream GBDT models to predict the 11 upstream targets and then concatenate their predictions to the downstream data features when training downstream GBDT models.

### 3.4 Hyperparameter Tuning

The standard hyperparameter tuning procedure for deep learning is to randomly sample a validation set and use it to optimize the hyperparameters. In contrast, in our scenario we often have too little downstream data to afford a sizeable validation split. However, we can make use of the abundant upstream data and leverage hyperparameter transfer which is known to be effective for GBDT (Woźnica et al., 2022; Grzyb et al., 2021).

We tune the hyperparameters of each model with the Optuna library (Akiba et al., 2019) using Bayesian optimization. In particular, for GBDT models and neural baselines trained from scratch, we tune the hyperparameters on a single randomly chosen upstream target with the same number of training samples as available in the downstream task, since hyperparameters depend strongly on the sample size. The optimal hyperparameters are chosen based on the upstream validation set, where validation data is plentiful. We find this tuning strategy to be especially effective for GBDT and provide comparison with default hyperparameters in Appendix C. The benefits of this hyperparameter tuning approach are less pronounced for deep baselines.

For deep models with transfer learning, we tune the hyperparameters on the full upstream data using the available large upstream validation set with the goal to obtain the best performing feature extractor for the pre-training multi-target task. We then fine-tune this feature extractor with a small learning rate on the downstream data. As this strategy offers considerable performance gains over default hyperparameters, we highlight the importance of tuning the feature extractor and present the comparison with default hyperparameters in Appendix C as well as the details on hyperparameter search spaces for each model.

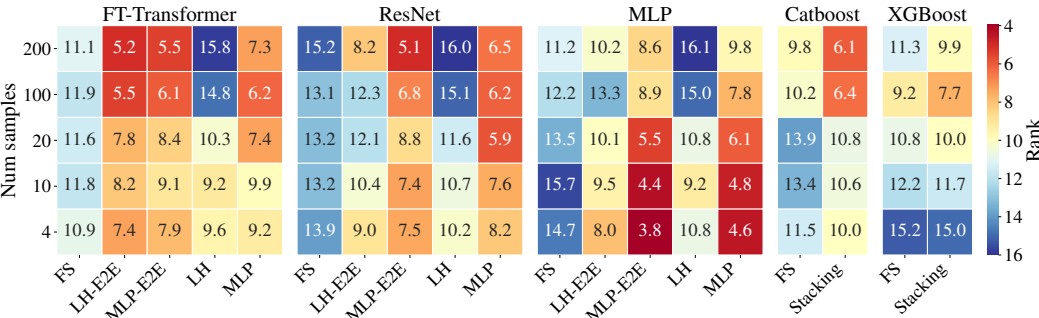

Figure 2: **Average model ranks across all downstream tasks.** Deep tabular models and GBDT performance is presented on the corresponding panels. Within each panel, columns represent transfer learning setups, and rows correspond to the number of available downstream samples. Warmer colors indicate better performance. FS denotes training from scratch (without pre-training on upstream data), LH and MLP denote linear and MLP heads correspondingly, E2E denotes end-to-end training. Rank is averaged across all downstream tasks. FT-Transformer fine-tuned end-to-end outperforms all GBDT models, including GBDT with stacking, at all data levels. MLP is highly competitive in low data regimes.

## 4 RESULTS FOR TRANSFER LEARNING WITH DEEP TABULAR MODELS

In this section, we compare transfer learned deep tabular models with GBDT methods at varying levels of downstream data availability. We note that here we present the aggregated results in the form of the average rank of the models across all of the twelve downstream tasks at each data level. We choose this rank aggregation metric since it does not allow a small number of high-variance tasks to dominate comparisons, unlike, for example, average accuracy. Ranks are computed taking into account statistical significance of the performance differences between the models. We further report the detailed results for all of the models on all datasets and results for TabTransformer in Appendix D.

Figure 2 presents average model ranks on the downstream tasks as a heatmap where the warmer colors represent better overall rank. The performance of every model is shown on the dedicated panel of the heatmap with the results for different transfer learning setups presented in columns. First, noting the color pattern in the Catboost and XGBoost columns, we observe that deep tabular models pre-trained on the upstream data outperform GBDT at all data levels and especially in the low data regime of 4-20 downstream samples.

We emphasize that knowledge transfer with stacking, while providing strong boosts compared to from-scratch GBDT training (see Stacking and FS columns of GBDT), still decisively falls behind the deep tabular models with transfer learning, suggesting that representation learning for tabular data is significantly more powerful and allows neural networks to transfer richer information than simple predictions learned on the upstream tasks.

We summarize the main practical takeaways of Figure 2 below:

- Simpler models such as MLP with transfer learning are competitive, especially in extremely low data regimes. More complex architectures like FT-Transformer offer consistent performance gains over GBDT across all data levels and reach their peak performance in higher data regimes.

- Representation learning with deep tabular models provides significant gains over strong GBDT baselines leveraging knowledge transfer from the upstream data through stacking. The gains are especially pronounced in low data regimes.

- Regarding transfer learning setups, we find that using an MLP head with a trainable or frozen feature extractor is effective for all deep tabular models. Additionally, a linear head with an end-to-end fine-tuned feature extractor is competitive for FT-Transformer.

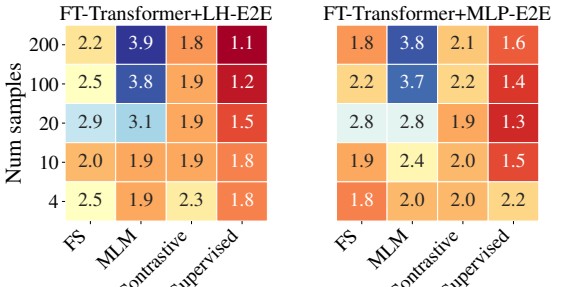

Figure 3: **Comparison of supervised and self-supervised pre-training strategies for FT-Transformer.** The left panel illustrates end-to-end fine-tuning with linear head and the right plot illustrates end-to-end fine-tuning with MLP head, the two most effective strategies for FT-Transformer. Within each panel, columns represent pre-training strategies and rows correspond to the number of available downstream samples. FS stands for training from scratch (without pre-training on upstream data), MLM and Contrastive denote masked-language-modeling-style and contrastive self-supervised pre-training strategies, and Supervised denotes supervised pre-training on upstream data. Warmer colors indicate better performance. Contrastive pre-training outperforms from-scratch trained models, while MLM pre-training is not effective. Supervised pre-training outperforms all self-supervised pre-training strategies in our experiments.

To verify that transfer learning with deep tabular models works well beyond the medical domain, we conduct experiments on other datasets: Yeast functional genomics data from the biological sciences domain (Elisseeff and Weston, 2001) and Emotions data from the music domain (Trohidis et al., 2008). Both datasets are multilabel, and we treat each classification label as a separate task. Similarly to the experiments with MetaMIMIC, we split the tasks into downstream and upstream by reserving $n - 1$ labels for the upstream task and the $n$-th label for the downstream task. We report results of these experiments in Figures 10, 11 in Appendix D.6. We observe similar trends and in particular that deep tabular models pre-trained on upstream data and finetuned with MLP head outperform deep baselines trained from scratch and Catboost models leveraging stacking.

## 5 Self-Supervised Pre-training

In domains where established SSL methods are increasingly dominant, such as computer vision, self-supervised learners are known to extract more transferable features than models trained on labelled data (He et al., 2020; 2021). In this section, we compare supervised pre-training with unsupervised pre-training and find that the opposite is true in the tabular domain. We use the Masked Language Model (MLM) pre-training recently adapted to tabular data (Huang et al., 2020) and the tabular version of contrastive learning (Somepalli et al., 2021). Since both methods were proposed for tabular transformer architectures, we conduct the experiments with the FT-Transformer model. The inferior performance of self-supervised pre-training might be a consequence of the fact that SSL is significantly less explored and tuned in the tabular domain than in vision or NLP.

### 5.1 Tabular MLM Pre-training

Masked Language Modeling (MLM) was first proposed for language models by Devlin et al. (2019) as a powerful unsupervised learning strategy. MLM involves training a model to predict tokens in text masked at random so that its learned representations contain information useful for reconstructing these masked tokens. In the tabular domain, instead of masking tokens, a random subset of features is masked for each sample, and the masked values are predicted in a multi-target classification manner (Huang et al., 2020). In our experiments, we mask one randomly selected feature for each sample, asking the network to learn the structure of the data and form representations from $n - 1$ features that are useful in producing the value in the $n$-th feature. For more detail, see Appendix B.

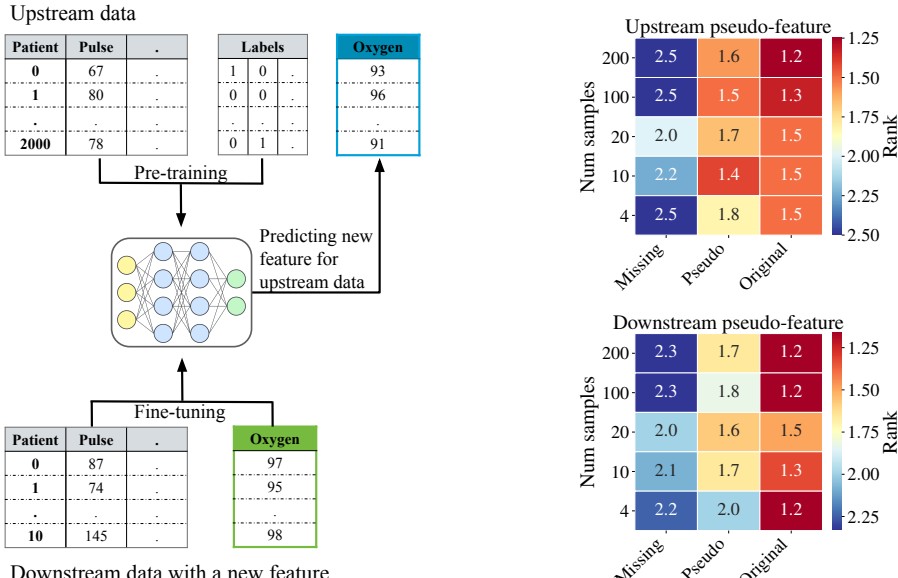

Figure 4: **Pseudo-Feature method for aligning upstream and downstream feature sets.** Left: Diagram illustrating strategy for handling mismatches between the upstream and downstream features. When upstream data is missing a feature present in downstream data, it is predicted with a model pre-trained on upstream data and fine-tuned to predict the new feature on the downstream data. When a feature is missing in the downstream data, it is predicted with a model trained to predict this feature on the upstream data. Right top: Scenario with missing feature in the upstream data. Comparison of ranks of FT-Transformer model trained on data with missing feature, with pseudo-feature and with the original feature. Right bottom: Scenario with missing feature in the downstream data. Comparison of ranks of FT-Transformer model trained and fine-tuned on data with missing feature, fine-tuned with pseudo and with original feature. In both scenarios, using the pseudo-feature is better than training without the feature but worse than worse than the original ground truth feature values.

## 5.2 CONTRASTIVE PRE-TRAINING

Contrastive pre-training uses data augmentations to generate positive pairs, or two different augmented views of a given example, and the loss function encourages a feature extractor to map positive pairs to similar features. Meanwhile, the network is also trained to map negative pairs, or augmented views of different base examples, far apart in feature space. We utilize the implementation of contrastive learning from Somepalli et al. (2021). In particular, we generate positive pairs by applying two data augmentations: CutMix (Yun et al., 2019) in the input space and Mixup (Zhang et al., 2017) in the embedding space. For more details, see Appendix B.

## 5.3 COMPARING SUPERVISED AND SELF-SUPERVISED PRE-TRAINING

While self-supervised learning makes for transferable feature extractors in other domains, our experiments show that supervised pre-training is consistently better than the recent SSL pre-training methods we try that are designed for tabular data. In Figure 3, we compare supervised pre-training with contrastive and MLM pre-training strategies and show that supervised pre-training always attains the best average rank. Contrastive pre-training produces better results than training from scratch on the downstream data when using a linear head, but it is still inferior to supervised pre-training. Tabular MLM pre-training also falls behind the supervised strategy and performs comparably to training from scratch in the lower data regimes but leads to a weaker downstream model in the higher data regimes.

## 6 Aligning Upstream and Downstream Feature Sets with Pseudo-Features

While so far we have worked with upstream and downstream tasks which shared a common feature space, in the real world, tabular data is highly heterogeneous and upstream data having a different set of features from downstream data is a realistic problem (Lewinson, 2020). In our medical data scenario, downstream patient data may include additional lab tests not available for patients in the upstream data. In fact, the additional downstream feature may not even be meaningful for the upstream data, such as medical exams only applicable to downstream patients of biological sex different from the upstream patients. In this section, we propose a pseudo-feature method for aligning the upstream and downstream data and show that the data heterogeneity problem can be addressed more effectively than simply taking the intersection of the upstream and downstream feature sets for tabular transfer learning, which would throw away useful features. The proposed pseudo-feature method is related to missing data imputation Zhang et al. (2008); Sefidian and Daneshpour (2019); Yoon et al. (2018) and self-training ideas Tanha et al. (2017). In particular, misalignment of feature sets in upstream and downstream data can be seen as an extreme scenario with features having all values missing.

Suppose our upstream data $(X_u, Y_u)$ is missing a feature $x_{new}$ present in the downstream data $(X_d, Y_d)$. We then use transfer learning in stages. As the diagram on the left panel of Figure 4 shows:

1. Pre-train feature extractor $f : X_u \to Y_u$ on upstream data $(X_u, Y_u)$ without feature $x_{new}$.

2. Fine-tune the feature extractor $f$ on the downstream samples $X_d$ to predict $x_{new}$ as the target and obtain a model $\hat{f} : X_d \setminus \{x_{new}\} \to x_{new}$.

3. Use the model $\hat{f}$ to assign pseudo-values $\hat{x}_{new}$ of the missing feature to the upstream samples: $\hat{x}_{new} = \hat{f}(X_u)$ thus obtaining augmented upstream data $(X_u \cup \{\hat{x}_{new}\}, Y_u)$.

4. Finally, we can leverage the augmented upstream data $(X_u \cup \{\hat{x}_{new}\}, Y_u)$ to pre-train a feature extractor which we will fine-tune on the original downstream task $(X_d, Y_d)$ using all of the available downstream features.

Similarly, in scenarios with a missing feature in downstream data, we can directly train a feature predictor on the upstream data and obtain pseudo-values for the downstream data.

This approach offers appreciable performance boosts over discarding the missing features and often performs comparably to models pre-trained with the ground truth feature values as shown in the right panel of Figure 4. The top heatmap represents the experiment where downstream data has an additional feature missing from the upstream data. The bottom heatmap represents the opposite scenario of the upstream data having an additional feature not available in the downstream data. To ensure that the features we experiment with are meaningful and contain useful information, for each task we chose important features according to GBDT feature importances. We observe that in both experiments, using the pseudo feature is better than doing transfer learning without the missing feature at all. Additionally, we observe that in some cases, the pseudo-feature approach performs comparably to using the original ground truth feature (10-100 samples on the top heatmap and 20 samples on the bottom heatmap). Interestingly, the pseudo-feature method is more beneficial when upstream features are missing, which suggests that having ground-truth values for the additional feature in the downstream data is more important.

## 7 Discussion

In this paper, we demonstrate that deep tabular models with transfer learning definitively outperform strong GBDT baselines with stacking in a realistic scenario where the target downstream data is limited and auxillary upstream pre-training data is available. We highlight that representation learning with neural networks enables more effective knowledge transfer than leveraging upstream task predictions through stacking. Additionally, we present a pseudo-feature method to enable effective transfer learning in challenging cases where the upstream and downstream feature sets differ. We provide suggestions regarding architectures, hyperparameter tuning, and setups for tabular transfer learning and hope that this work serves as a guide for practitioners.

## 8 ETHICS STATEMENT

We conduct experiments with the MetaMIMIC medical dataset (Grzyb et al., 2021; Woźnica et al., 2022), which is based on the publicly accessible MIMIC database (Goldberger et al., 2000; Johnson et al., 2016; 2021). Regarding the patient consent to collect this data, as stated in (Johnson et al., 2016): "The project was approved by the Institutional Review Boards of Beth Israel Deaconess Medical Center (Boston, MA) and the Massachusetts Institute of Technology (Cambridge, MA). Requirement for individual patient consent was waived because the project did not impact clinical care and all protected health information was deidentified". The MIMIC database is freely available to any person upon completion of the credentialing process found at the following link: `https://mimic-iv.mit.edu/docs/access/` and is distributed under Open Data Commons Open Database License v1.0, please see the following link for details: `https://physionet.org/content/mimic-iv-demo-omop/view-license/0.9/`.

## 9 REPRODUCIBILITY STATEMENT

We include the code for reproducing our results in the supplementary materials. We also describe strategies for data preprocessing, training the models, choosing the best epoch as well as hyperparameter ranges in Appendix B and C.

## 10 ACKNOWLEDGMENTS

This work was supported by the Office of Naval Research, the National Science Foundation (IIS-2212182), and Capital One Bank.

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

## A    LIMITATIONS AND IMPACT

In this section we discuss limitations of our work. We note that transfer learning with deep tabular models requires more computational resources than training XGBoost or CatBoost, especially when the hyperparameter tuning is used thus incurring additional costs on practitioners adopting our approach. In addition, while we handle the cases of differing upstream and downstream feature sets thus allowing for transfer learning with heterogeneous data, our approach relies on having reasonably similar (but not identical) upstream and downstream feature sets and related upstream and downstream tasks and we do not propose a foundation model for tabular data.

## B    EXPERIMENTAL DETAILS.

### B.1    HARDWARE

We ran our experiments on NVIDIA GeForce RTX 2080 Ti machines.

### B.2    IMPLEMENTATION LICENSES

For the model implementations and training routines we adapt the code from the following publicly available repositories and libraries:

- RTDL [1] under MIT License
- TabTransformer [2] under MIT License
- Catboost [3] under Apache License
- XGBoost [4] under Apache License

### B.3    DETAILS ON METAMIMIC TARGETS

The MetaMIMIC Repository (Grzyb et al., 2021; Woźnica et al., 2022) is based on the publicly accessible MIMIC database (Goldberger et al., 2000; Johnson et al., 2016; 2021) which contains health related data from patients admitted to the ICU. MetaMIMIC contains health records for 34925 unique patients, who are at least 15 years old, were diagnosed with at least one of considered diseases, and did not stay in hospital for more than 60 days. The target diagnoses correspond to the 50 most common conditions, which are also found in both ICD-9 and ICD-10 codes (International Classification of Diseases), which results in 12 unique diagnoses. Please, see Table 1 for selected targets, their ICD codes and frequency in the considered cohort (adapted from Table 1 in Woźnica et al. (2022)).

To understand how related the MetaMIMIC tasks are, we compute pair-wise correlations between targets and present them as a heatmap in Figure 5. We find that the maximum correlation of 0.36 is between ischematic and hypertensive diagnoses, while the average absolute pair-wise correlation is 0.08. We conclude that although certain tasks are slightly correlated, such as targets corresponding to cardiovascular diseases, the average pair-wise correlation between all the targets is low.

### B.4    DETAILS ON THE UPSTREAM-DOWNSTREAM SPLITS AND THE NUMBER OF SEEDS

We reserve 6985 patients (20% of the MetaMIMIC data) for the downstream test set, and use 22701 patients (65% of the MetaMIMIC data) for training and 5239 patients (15% of the MetaMIMIC data) – as a validation set for hyperparameter tuning of the upstream feature extractors. We further simulate scenarios with different levels of data availability by sampling from the training data downstream patients for fine-tuning. We note that these splits do not change across seeds. We use similar splits on Yeast and Emotions datasets.

---

[1] https://github.com/Yura52/tabular-dl-revisiting-models
[2] https://github.com/lucidrains/tab-transformer-pytorch
[3] https://catboost.ai/
[4] https://xgboost.ai/

| Condition | ICD-9 | ICD-10 | Frequency |
|---|---|---|---|
| Hypertensive diseases | 401-405 | I10-I16 | 59.8% |
| Disorders of lipoid metabolism | 272 | E78 | 40.3% |
| Anemia | 280-285 | D60-D64 | 35.9% |
| Ischematic heart disease | 410-414 | I20-I25 | 32.8% |
| Diabetes | 249-250 | E08-E13 | 25.3% |
| Chronic lower respiratory diseases | 466, 490-496 | J40-J47 | 19.5% |
| Heart failure | 428 | I50 | 19.4% |
| Hypotension | 458 | I95 | 14.5% |
| Purpura and other hemorrhagic conditions | 287 | D69 | 11.9% |
| Atrial fibrillation and flutter | 427.3 | I48 | 10.5% |
| Overweight, obesity and other hyperalimentation | 278 | E65-E68 | 10.5% |
| Alcohol dependence | 303 | F10 | 7.7% |

Table 1: Selected targets with corresponding ICD codes and frequency in the considered cohort.

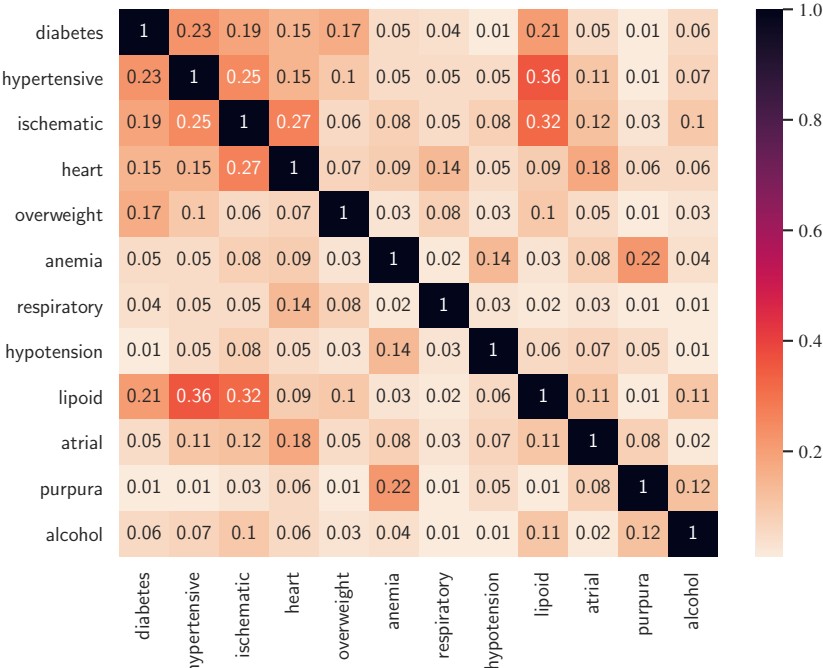

Figure 5: **Correlations between MetaMIMIC targets.** The heatmap represents absolute pair-wise correlations between MetaMIMIC targets.

We run each experiment with 10 random seeds for the experiments on transfer learning with deep tabular models in Section 4 and Appendix D, and with 5 random seeds for the experiments with self-supervised pre-training and feature imputation in Sections 5, 6 and Appendix D.

## B.5 DATA PREPROCESSING

We preprocess numerical features with quantile transformation with standard output distribution for neural networks and we use original features for GBDT models. Quantile transformer is first fit on upstream data and then applied to downstream data to preserve similar feature distribution in upstream and downstream data. We note, that using the same normalization for upstream and downstream data is a very important step for transfer learning. We impute missing values with mean values for numerical features and with a new category for categorical features.

### B.6 Training Details

#### B.6.1 Supervised Pre-training

All deep models are trained with AdamW optimizer (Loshchilov and Hutter, 2017). We pre-train models on upstream data for 500 epochs with patience set to 30, meaning that we continue training until there are 30 consecutive epochs without improvement on the validation set, but no longer than 500 epochs. Since we assume limited data availability for downstream task, we do not sample a validation set for early stopping. Instead, we fine-tune/train models from scratch for 200 epochs and choose an "optimal" epoch as discussed in Section B.6.3. We use learning rate $1e - 4$ for training from scratch on downstream data and learning rate $5e - 5$ for fine-tuning pre-trained models. For pre-training, learning rate and weight decay are tunable hyperparameters for each model and each pre-training dataset (collection of $n - 1$ upstream tasks). Batch size was set to 256 in all transfer-learning experiments.

#### B.6.2 Self-supervised Pre-training

**MLM Pre-training**

To implement MLM pre-training for tabular data, for each training sample in a batch we randomly select a feature to replace with a masking token (in the embedding space), which is unique for every feature. We also initialize a distinct fully connected layer, or head, for each column which is used to predict the masked values of that feature. We compute the appropriate loss, cross-entropy for categorical features and MSE for numerical ones, using the output from the head corresponding to the masked feature.

Our masked pre-training routine has very no additional hyperparameters. The only deviations form the pre-training hyperparameters listed above are that we use larger batch sizes of 512 and we do not employ patience. In this setting we pre-training for a full 500 epochs.

Since MLM pre-training requires training additional classification heads for every feature in the data, it dramatically increases the memory requirements. Because of that we limit the experiments with self-supervised pre-training to using the default FT-transformer configuration, including the setups we compare to, that is training from scratch and using supervised pre-training.

**Contrastive Learning**

We adapt the implementation of contrastive learning for tabular data from Somepalli et al. (2021). The pre-training loss consists of two components, the first is the InfoNCE contrastive loss, which minimizes the distance between the representation of two views of the same data point (original and augmented samples), while maximizing the distance in the feature space between different samples. The second component is denoising loss, which is used to train an MLP head to predict the original sample from a noisy view of that sample. For a detailed explanation of contrastive pre-training for tabular data we refer the reader to Somepalli et al. (2021). To construct positive pairs for contrastive loss and noisy samples for denoising loss we use two data aumentations: CutMix in the input space and Mixup in the embedding space Yun et al. (2019); Zhang et al. (2017).

Formally, let a batch of data be represented by $X = \{x_i\}_{i=0}^n$, where each $x_i$ has $q$ features. Let $m$ denote a binary mask (over the features of any $x_i$) with each entry being a one with probability $p$. Then CutMix augmentation of a sample in the input space is computed as

$$x_i^{\text{CutMix}} = m \times x_i + (1 - m) \times x_j$$

where $j$ is a random index chosen from $[0, n]$. To define Mixup in the embedding space, let $\hat{X}^{\text{CutMix}}$ be the set of the embeddings of the entire CutMix-ed batch. Then, Mixup augmentation is computed as a convex combination

$$\hat{x}_i^{\text{Mixup}} = \mu \hat{x}_i^{\text{CutMix}} + (1 - \mu)\hat{x}_k^{\text{CutMix}}$$

where $k$ is again a random index chosen from $[0, n]$.

The hyperparameters are similar to supervised pre-training with only several modifications. First, we use smaller batch sized of 200 and we train for a full 500 epochs (no patience). Also, contrastive learning has additional hyperparameters $m$ and $\mu$, both of which we set to 0.9.

### B.6.3 EPOCH SELECTION

Since we work with limited downstream data, sampling a validation set for early stopping is not always possible. Therefore, we select the number of fine-tuning epochs for deep models as follows. For the data levels of 4 and 10 samples, we simply select 30 fine-tuning epochs. For more data of 20 samples, we select 60 fine-tuning epochs. In the larger data levels of 100 and 200 samples, we sample 20% of the data as a validation set to perform early stopping with the flexible end-to-end fine-tuned transfer learning setups prone to overfitting. For early stopping, we terminates training if no improvement in the validation score is observed for more than 30 epochs. In the less flexible transfer learning setups with a frozen feature extractor, we select 100 fine-tuning epochs for the MLP head atop a frozen feature extractor and 200 fine-tuning epochs for the linear head atop a frozen feature extractor. Finally, for the deep baselines with the hyperparameters tuned on a small subsample of the upstream data, we select the best epoch from the small upstream subsample.

### B.6.4 STACKING FOR GBDT MODELS

For XGBoost models we incorporate stacking by training 11 additional XGBoost classifiers on upstream tasks and stack their predictions as additional features for downstream data. For Catboost we apply the same strategy, but we train a single multi-label Catboost classifier predicting all 11 upstream labels.

### B.6.5 STATISTICAL SIGNIFICANCE

We compute ranks taking into account statistical significance of the performance differences between the models; on a given task, top models without statistically significant performance difference all receive rank 1. To compute statistical significance we use the one-sided Wilcoxon Rank-Sum test Wilcoxon (1945); Mann and Whitney (1947) with $p = 0.05$. We run each experiment with 10 random seeds for the experiments in Section 4 and 5 random seeds for the experiments in Sections 5, 6.

## C   HYPERPARAMETER TUNING

In this section we provide hyperparaeter search spaces and distributions for Optuna for each model, which are adapted from the original papers. For Catboost and XGBoost models we use search spaces proposed in Gorishniy et al. (2021). We run 50 Optuna trials to tune hyperparameters for each model. In our experiments we tune the hyperparameters on full upstream data for deep tabular models with transfer learning, and on upstream data of the same size as downstream data for deep baselines trained from scratch and for GBDT models.

### C.0.1   FT-TRANSFORMER

The hyperparameter search space and distributions as well as the default configuration are presented in Table 2. The number of heads is always set to 8 as recommended in the original paper.

### C.0.2   RESNET

The hyperparameter search space and distributions as well as the default configuration are presented in Table 3

### C.0.3   MLP

The hyperparameter search space and distributions as well as the default configuration are presented in Table 4

Table 2: Optuna hyperparameter search space and default configuration for FT-Transformer

| Parameter | Search Space | Default |
|---|---|---|
| Number of layers | UniformInt$[1, 4]$ | 3 |
| Feature embedding size | UniformInt$[64, 512]$ | 192 |
| Residual dropout | $\{0, \text{Uniform}[0, 0.2]\}$ | 0.0 |
| Attention dropout | Uniform$[0, 0.5]$ | 0.2 |
| FFN dropout | Uniform$[0, 0.5]$ | 0.1 |
| FFN factor | Uniform$[2/3, 8/3]$ | $4/3$ |
| Learning rate | LogUniform$[1e-5, 1e-3]$ | $1e-4$ |
| Weight decay | LogUniform$[1e-6, 1e-3]$ | $1e-5$ |

Table 3: Optuna hyperparameter search space and default configuration for ResNet

| Parameter | Search Space | Default |
|---|---|---|
| Number of layers | UniformInt$[1, 8]$ | 5 |
| Category embedding size | UniformInt$[64, 512]$ | 128 |
| Layer size | UniformInt$[64, 512]$ | 200 |
| Hidden factor | Uniform$[1, 4]$ | 3 |
| Hidden dropout | Uniform$[0, 0.5]$ | 0.2 |
| Residual dropout | $\{0, \text{Uniform}[0, 0.5]\}$ | 0.2 |
| Learning rate | LogUniform$[1e-5, 1e-2]$ | $1e-4$ |
| Weight decay | $\{0, \text{LogUniform}[1e-6, 1e-3]\}$ | 0.0 |

Table 4: Optuna hyperparameter search space and default configuration for MLP

| Parameter | Search Space | Default |
|---|---|---|
| Number of layers | UniformInt$[1, 8]$ | 3 |
| Category embedding size | UniformInt$[64, 512]$ | 128 |
| Layer size | UniformInt$[1, 512]$ | $[300, 200, 300]$ |
| Dropout | $\{0, \text{Uniform}[0, 0.5]\}$ | 0.2 |
| Learning rate | LogUniform$[1e-5, 1e-2]$ | $1e-4$ |
| Weight decay | $\{0, \text{LogUniform}[1e-6, 1e-3]\}$ | $1e-5$ |

### C.0.4 TABTRANSFORMER

The hyperparameter search space and distributions as well as the default configuration are presented in Table 5.

Table 5: Optuna hyperparameter search space and default configuration for TabTransformer

| Parameter | Search Space | Default |
|---|---|---|
| Number of heads | UniformInt$[2, 8]$ | 8 |
| Number of layers | UniformInt$[1, 12]$ | 6 |
| Category embedding size | UniformInt$[8, 128]$ | 32 |
| Attention Dropout | Uniform$[0.0, 0.5]$ | 0.0 |
| FF Dropout | Uniform$[0.0, 0.5]$ | 0.0 |
| Learning rate | LogUniform$[1e-6, 1e-3]$ | $1e-4$ |
| Weight decay | LogUniform$[1e-6, 1e-3]$ | $1e-5$ |

### C.0.5 CATBOOST

The hyperparameter search space and distributions are presented in Table 6. For default configuration we use default parameters from the Catboost library (Prokhorenkova et al., 2018).

Table 6: Optuna hyperparameter search space for Catboost

| Parameter | Search Space |
|---|---|
| Iterations | UniformInt$[2, 1000]$ |
| Max depth | UniformInt$[3, 10]$ |
| Learning rate | LogUniform$[1e-5, 1]$ |
| Bagging temperature | Uniform$[0, 1]$ |
| L2 leaf reg | LogUniform$[1, 10]$ |
| Leaf estimation iterations | UniformInt$[1, 10]$ |

### C.0.6 XGBOOST

The hyperparameter search space and distributions as well as the default configuration are presented in Table 7. For default configuration we use default parameters from the XGBoost library (Chen and Guestrin, 2016).

Table 7: Optuna hyperparameter search space for XGBoost

| Parameter | Search Space |
|---|---|
| Num estimators | UniformInt$[2, 1000]$ |
| Max depth | UniformInt$[3, 10]$ |
| Min child weight | LogUniform$[1e-8, 1e5]$ |
| Subsample | Uniform$[0.5, 1]$ |
| Learning rate | LogUniform$[1e-5, 1]$ |
| Col sample by level | Uniform$[0.5, 1]$ |
| Col sample by tree | Uniform$[0.5, 1]$ |
| Gamma | $\{0, \text{LogUniform}[1e-8, 1e2]\}$ |
| Lambda | $\{0, \text{LogUniform}[1e-8, 1e2]\}$ |
| Alpha | $\{0, \text{LogUniform}[1e-8, 1e2]\}$ |

### C.1 TUNING GBDT MODELS

In Table 8 we explore the effectiveness of our hyperparameter tuning strategy for GBDT models. In particular, we compute average ranks for models with tuned and default configurations. Recall, because of the limited data availability in downstream tasks, we tune the hyperparameters on upstream data of the same size as the downstream data using full-size validation set. We find that using upstream data for tuning hyperparameters is effective for XGBoost model while for catboost tuned configuration outperforms default configuration at three out of five data availability levels.

Table 8: **Comparison of tuned and default configurations of GBDT models.** The table displays ranks computed pair-wise for default/tuned configurations of XGBoost and Catboost models.

| Num samples | 4 | 10 | 20 | 100 | 200 |
|---|---|---|---|---|---|
| XGBoost Tuned | 1.17 | 1.17 | 1.25 | 1.33 | 1.42 |
| XGBoost Default | 1.42 | 1.83 | 1.67 | 1.50 | 1.58 |
| Catboost Tuned | 1.33 | 1.08 | 1.17 | 1.25 | 1.42 |
| Catboost Default | 1.25 | 1.25 | 1.42 | 1.33 | 1.33 |

### C.2 TUNING DEEP BASELINES

In Table 9 we evaluate hyperparameter tuning routine for deep baselines, i.e. deep neural networks trained from scratch. We find that unlike GBDT models, hyperparameters tuned on upstream data do not transfer to downstream tasks at lower data regimes (i.e. 4-20 samples) for deep models. However, tuning helps deep baselines in higher data regimes.

Table 9: **Comparison of tuned and default configurations of deep tabular baselines.** The table displays ranks computed pair-wise for default/tuned configurations of deep tabular neural networks.

| Num samples | 4 | 10 | 20 | 100 | 200 |
|---|---|---|---|---|---|
| FT-Transformer Tuned | 1.33 | 1.25 | 1.00 | 1.33 | 1.33 |
| FT-Transformer Default | 1.00 | 1.08 | 1.58 | 1.417 | 1.50 |
| ResNet Tuned | 1.25 | 1.17 | 1.33 | 1.25 | 1.75 |
| ResNet Default | 1.00 | 1.08 | 1.25 | 1.00 | 1.00 |
| MLP Tuned | 1.42 | 1.67 | 1.58 | 1.17 | 1.33 |
| MLP Default | 1.00 | 1.08 | 1.33 | 1.67 | 1.50 |
| TabTransformer Tuned | 1.25 | 1.33 | 1.17 | 1.08 | 1.17 |
| TabTransformer Default | 1.17 | 1.25 | 1.33 | 1.50 | 1.67 |

## C.3 TUNING DEEP MODELS FOR TRANSFER LEARNING

In Table 10 we evaluate the hyperparameter tuning strategy for deep tabular neural networks with transfer learning. Recall, we tune the hyperparameters on the full upstream data which is then used for pre-training. We find this strategy helpful for most models, especially for FT-Transformer and TabTransformer, and for most transfer learning setups.

Table 10: **Comparison of tuned and default configurations of deep tabular neural networks with transfer learning.** The table displays ranks computed pair-wise for default/tuned configurations of deep tabular models fine-tuned with 4 different transfer-learning setups.

| Num Samples | 4 | 10 | 20 | 100 | 200 |
|---|---|---|---|---|---|
| FT-Transformer + LH-E2E Tuned | 1.00 | 1.00 | 1.00 | 1.08 | 1.17 |
| FT-Transformer + LH-E2E Default | 1.25 | 1.25 | 1.17 | 1.42 | 1.17 |
| FT-Transformer + MLP-E2E Tuned | 1.00 | 1.00 | 1.08 | 1.00 | 1.08 |
| FT-Transformer + MLP-E2E Default | 1.08 | 1.00 | 1.25 | 1.33 | 1.58 |
| FT-Transformer + LH Tuned | 1.00 | 1.00 | 1.00 | 1.00 | 1.00 |
| FT-Transformer + LH Default | 1.17 | 1.25 | 1.33 | 1.58 | 1.67 |
| FT-Transformer + MLP Tuned | 1.08 | 1.00 | 1.00 | 1.00 | 1.00 |
| FT-Transformer + MLP Default | 1.17 | 1.08 | 1.33 | 1.58 | 1.75 |
| ResNet + LH-E2E Tuned | 1.00 | 1.17 | 1.33 | 1.25 | 1.33 |
| ResNet + LH-E2E Default | 1.08 | 1.00 | 1.00 | 1.17 | 1.25 |
| ResNet + MLP-E2E Tuned | 1.00 | 1.00 | 1.08 | 1.08 | 1.00 |
| ResNet + MLP-E2E Default | 1.17 | 1.00 | 1.08 | 1.58 | 1.42 |
| ResNet + LH Tuned | 1.00 | 1.00 | 1.08 | 1.00 | 1.00 |
| ResNet + LH Default | 1.17 | 1.25 | 1.25 | 1.50 | 1.50 |
| ResNet + MLP Tuned | 1.00 | 1.00 | 1.00 | 1.00 | 1.08 |
| ResNet + MLP Default | 1.17 | 1.17 | 1.33 | 1.58 | 1.75 |
| MLP + LH-E2E Tuned | 1.17 | 1.25 | 1.42 | 1.42 | 1.17 |
| MLP + LH-E2E Default | 1.08 | 1.08 | 1.08 | 1.33 | 1.50 |
| MLP + MLP-E2E Tuned | 1.00 | 1.08 | 1.33 | 1.25 | 1.42 |
| MLP + MLP-E2E Default | 1.08 | 1.00 | 1.25 | 1.25 | 1.33 |
| MLP + LH Tuned | 1.00 | 1.25 | 1.17 | 1.17 | 1.08 |
| MLP + LH Default | 1.08 | 1.08 | 1.17 | 1.17 | 1.25 |
| MLP + MLP Tuned | 1.08 | 1.08 | 1.08 | 1.00 | 1.00 |
| MLP + MLP Default | 1.00 | 1.00 | 1.17 | 1.17 | 1.42 |
| TabTransformer + LH-E2E Tuned | 1.00 | 1.00 | 1.08 | 1.08 | 1.00 |
| TabTransformer + LH-E2E Default | 1.17 | 1.25 | 1.25 | 1.50 | 1.25 |
| TabTransformer + MLP-E2E Tuned | 1.00 | 1.00 | 1.08 | 1.08 | 1.00 |
| TabTransformer + MLP-E2E Default | 1.17 | 1.25 | 1.25 | 1.42 | 1.25 |
| TabTransformer + LH Tuned | 1.00 | 1.00 | 1.00 | 1.00 | 1.00 |
| TabTransformer + LH Default | 1.25 | 1.17 | 1.25 | 1.25 | 1.17 |
| TabTransformer + MLP Tuned | 1.00 | 1.00 | 1.00 | 1.00 | 1.00 |
| TabTransformer + MLP Default | 1.25 | 1.33 | 1.25 | 1.25 | 1.25 |

# D    ADDITIONAL RESULTS

## D.1    RESULTS FOR TABTRANSFORMER

Figure 6 is equivalent to Figure 2, but also includes the results for TabTransformer model, a tabular neural network consisting of transformer block for categorical features and an MLP block on top for numerical features. We find that TabTransformer performs poorly compared to other deep tabular neural networks, which might be explained by the fact that the original paper Huang et al. (2020) only suggests to tune the hyperparameters of transformer block, but not the MLP part, while Meta-MIMIC data has only one categorical features and 171 numerical features.

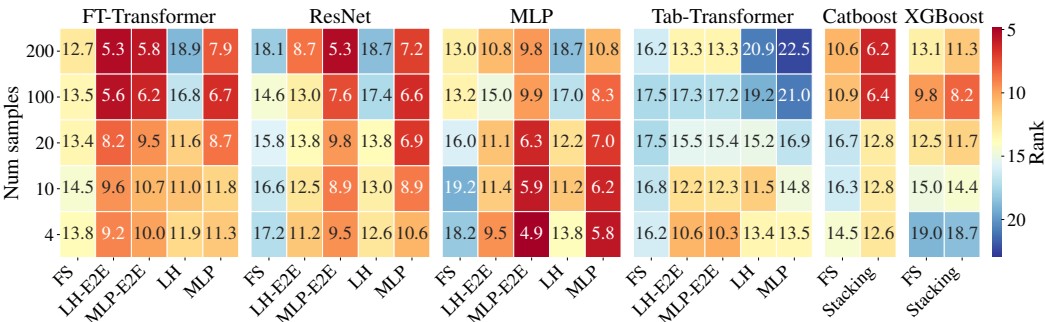

Figure 6: **Average model ranks including TabTransformer.** Deep tabular models and GBDT performance is presented on the corresponding panels. Within each panel, columns represent transfer learning setups, and rows correspond to the number of available downstream samples. Warmer colors indicate better performance. FS denotes training from scratch (without pre-training on upstream data), LH and MLP denote linear and MLP heads correspondingly, E2E denotes end-to-end training. Rank is averaged across all downstream tasks.

## D.2    RANKS STANDARD ERROR

Figure 7 presents standard errors for ranks in Figure 2 computed across tasks.

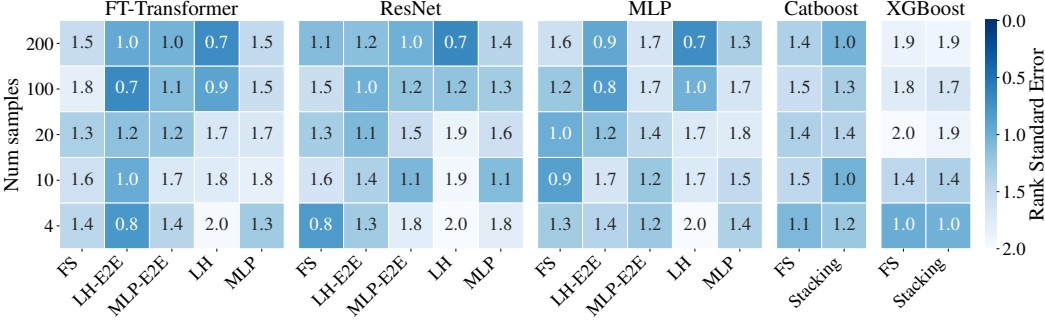

Figure 7: **Standard error for model ranks computed across all downstream tasks.** This figure complements Figure 2. Darker colors indicate lower standard errors.

## D.3    MODEL-WISE RANKS

In Figure 8 we compare different transfer learning setups for each deep tabular model.

## D.4    AVERAGE ROC-AUC IMPROVEMENT OVER CATBOOST

In Figure 9 we display ROC-AUC improvements over Catboost baseline averaged across all downstream tasks. We observe trends similar to ones with ranks; MLP model performs best in low data regimes, while FT-Transformer offers consistent gains across all data levels.

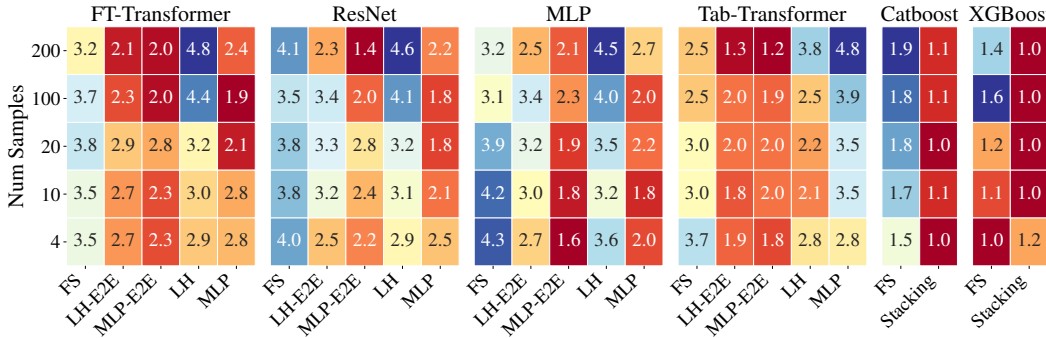

Figure 8: **Model-wise ranks for GBDT and neural networks.** Within each panel, columns represent transfer learning setups, and rows correspond to the number of available downstream samples. Warmer colors indicate better performance. FS denotes training from scratch (without pre-training on upstream data), LH and MLP denote linear and MLP heads correspondingly, E2E denotes end-to-end training. Rank is computed across training setups for each model and is averaged across all downstream tasks.

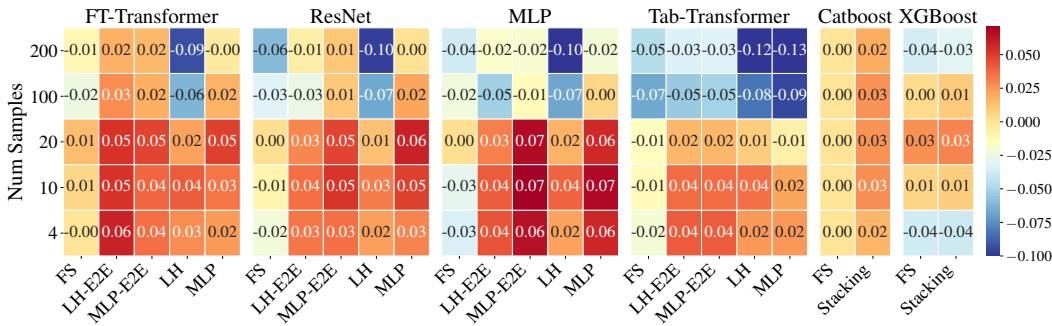

Figure 9: **Average ROC-AUC improvements over Catboost baseline.** Within each panel, columns represent transfer learning setups, and rows correspond to the number of available downstream samples. Warmer colors indicate better performance. FS denotes training from scratch (without pre-training on upstream data), LH and MLP denote linear and MLP heads correspondingly, E2E denotes end-to-end training. ROC-AUC improvement is computed as difference in ROC-AUC with Catboost model and is averaged across all downstream tasks.

### D.5 DATASET-LEVEL RESULTS

In Tables 11, 12, 13, 14 we report ROC-AUC measurements for each model on each downstream task. We include the results for GBDT models, neural baselines and neural networks with transfer learning.

### D.6 EXPERIMENTS ON ADDITIONAL DATASETS

To expand the experimental evaluation of tabular transfer learning beyond the medical domain, we include experiments on two additional multi-label datasets: Yeast functional genomics data Elisseeff and Weston (2001) and Emotions data Trohidis et al. (2008) . Yeast contains micro-array expression data and phylogenetic profiles of genes with labels corresponding to different gene functional classes. Each gene is associated with a set of functional classes, therefore each task aims to predict if a gene is associated with a particular gene functional group. The dataset contains 2417 samples (genes) with 103 numerical features and 14 labels for each sample. Emotions data aims to detect emotions in music, it contains rhythmic and timbre features of 593 songs and labels corresponding to 6 emotions. In our experimental setup, we treat each classification label as a separate task. Similarly to the experiments with MetaMIMIC, we split the tasks into downstream and upstream by reserving $n-1$ labels for the upstream task and the $n$-th label for the downstream task. We further limit the amount of downstream data to 4, 10, 20, 100 and 200 samples. This strategy results in 70 combinations of

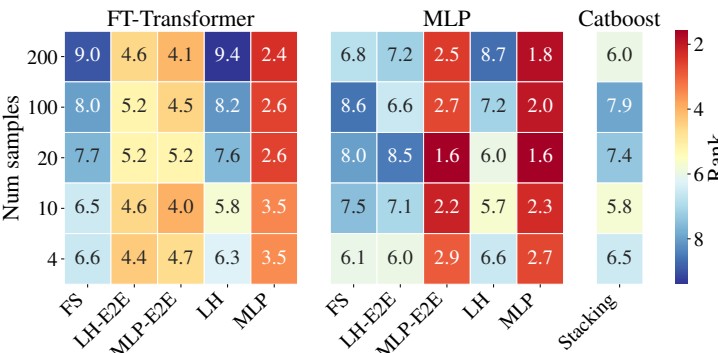

Figure 10: **Average model ranks across downstream tasks in Yeast data.** Deep tabular models and Catboost performance is presented on the corresponding panels. Within each panel, columns represent transfer learning setups, and rows correspond to the number of available downstream samples. Warmer colors indicate better performance. FS denotes training from scratch (without pre-training on upstream data), LH and MLP denote linear and MLP heads correspondingly, E2E denotes end-to-end training. Rank is averaged across all downstream tasks.

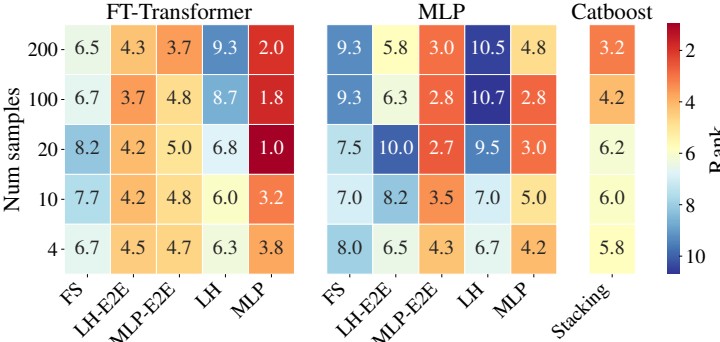

Figure 11: **Average model ranks across downstream tasks in Emotions data.** Deep tabular models and Catboost performance is presented on the corresponding panels. Within each panel, columns represent transfer learning setups, and rows correspond to the number of available downstream samples. Warmer colors indicate better performance. FS denotes training from scratch (without pre-training on upstream data), LH and MLP denote linear and MLP heads correspondingly, E2E denotes end-to-end training. Rank is averaged across all downstream tasks.

upstream and downstream datasets constructed from the Yeast database and 30 combinations from the Emotions database. We conduct experiments with FT-Transformer, MLP, and Catboost models which demonstrated the best results in our previous experiments. We tune the hyperparameters for deep models and Catboost as described in Appendix C. Figures 10, 11 present average model ranks of two tabular neural network and Catboost models on downstream tasks as a heatmap with warmer colors representing better results.

We observe that deep tabular models pre-trained on upstream data and finetuned with MLP head outperform models trained from scratch and Catboost models leveraging stacking. Interestingly, MLP model performs better on the Yeast dataset, while FT-Transformer performs better on the Emotions dataset.

## D.7    EXPERIMENTS WITH FEWER PRE-TRAINING TASKS

To further validate the effectiveness of transfer learning in scenarios where fewer pre-training tasks are available we repeat our experiments with the first 5 targets from Meta-MIMIC dataset: Diabetes,

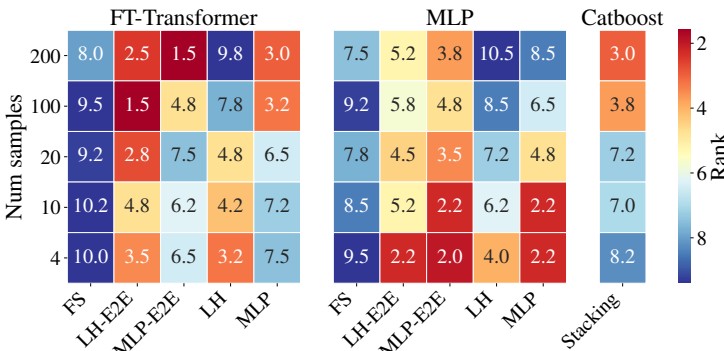

Figure 12: **Average model ranks for models pre-trained and finetuned on the first five MetaMIMIC targets.** Deep tabular models and Catboost performance is presented on the corresponding panels. Within each panel, columns represent transfer learning setups, and rows correspond to the number of available downstream samples. Warmer colors indicate better performance. FS denotes training from scratch (without pre-training on upstream data), LH and MLP denote linear and MLP heads correspondingly, E2E denotes end-to-end training. Rank is averaged across 5 downstream tasks.

Hypertensive, Ischematic, Heart, Overweight. In particular we pre-train on 4 available targets and finetune on the remaining task. We limit the number of downstream tasks to 5 for consistency of the number of pre-training targets across downstream tasks. Figure 12 presents the results of this experiment, we observe similar trends to the original experiments with all targets. In particular, we find that deep neural networks pre-trained even on fewer upstream targets definitively outperform Catboost with stacking with MLP model performing better in low-data regimes and FT-Transformer performing better when more downstream data is available.

## D.8 WHEN IS TRANSFER LEARNING MOST EFFECTIVE? CASE STUDY

One might wonder when deep neural networks are effective for transferring knowledge between tasks. We hypothesize that transfer learning is most effective in scenarios where upstream and downstream tasks are related. In the previous experiments with pre-training on fewer tasks (see Appendix D.7) we found that transfer learning is effective when transferring knowledge between the first five tasks, which include a few cardiovascular-related conditions (*hypertensive, ischematic, heart*), *diabetes* and *overweight* conditions. From the correlation matrix in Figure 5, we can see that these tasks are slightly more correlated with each other than with other conditions such as *anemia, respiratory, hypotension, purpura, alcohol*. We notice that the latter set of tasks is also less related to cardiovascular conditions. To evaluate our hypothesis that transfer learning is less effective when pre-training and downstream tasks are less related, we run experiments with pre-training the model on the first 5 related tasks and transferring knowledge to the set of "less related" tasks. The results for this experiment are presented in Figure 13, we observe that although pre-trained MLP model still provides performance boosts in small data regimes compared to Catboost, the trends are less pronounced compared to results for transferring knowledge to "more related tasks" in Figure 12.

## D.9 EXPERIMENTS WITH HIGHER DATA REGIMES IN DOWNSTREAM DATA

To expand the experimental evaluation to higher data regimes in downstream data we repeat our experiments with 400, 1000 and 2000 samples available for finetuning in the downstream data. The results for this experiment are presented in Figure 14. We find that even in higher data regimes pre-training on upstream data provides significant boost to FT-Transformer model compared to Catboost with stacking. At the same time, MLP performs on par with Catboost when there is more downstream data available, which aligns with our previous results.

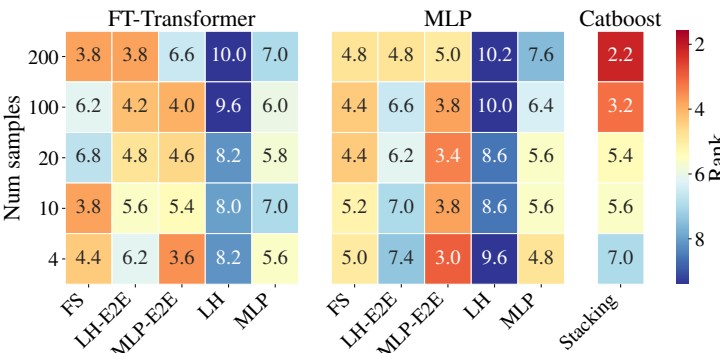

Figure 13: **Average model ranks for transferring knowledge to "less related" tasks.** Deep tabular models and Catboost performance is presented on the corresponding panels. Within each panel, columns represent transfer learning setups, and rows correspond to the number of available downstream samples. Warmer colors indicate better performance. FS denotes training from scratch (without pre-training on upstream data), LH and MLP denote linear and MLP heads correspondingly, E2E denotes end-to-end training. Rank is averaged across 5 "less related tasks".

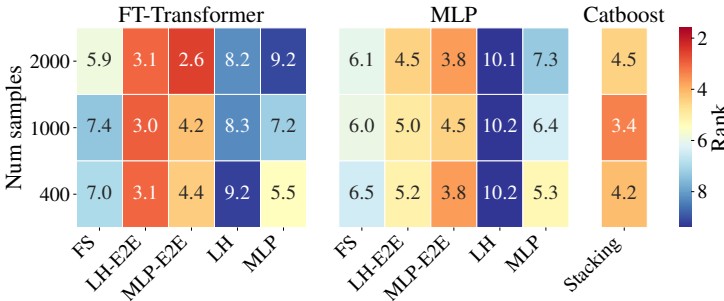

Figure 14: **Average model ranks in higher-data regimes for downstream tasks.** Deep tabular models and Catboost performance is presented on the corresponding panels. Within each panel, columns represent transfer learning setups, and rows correspond to the number of available downstream samples. Warmer colors indicate better performance. FS denotes training from scratch (without pre-training on upstream data), LH and MLP denote linear and MLP heads correspondingly, E2E denotes end-to-end training. Rank is averaged across all downstream tasks.

### D.10 ADDITIONAL EXPERIMENTS WITH PSEUDO-FEATURE METHOD

We further verify the effectiveness of our pseudo-feature method in scenarios with multiple features misaligned between upstream and downstream data. We repeat the experiment from Section 6, but instead of a single feature we remove three features. In Figure 15, the top left heatmap represents the experiment where the downstream data has three additional features missing from the upstream data. The bottom left heatmap represents the opposite scenario of the upstream data having three additional features not available in the downstream data. To ensure that the features we experiment with are meaningful and contain useful information, we chose important features according to GBDT feature importances. In Figure 15, we include the average ranks of FT-Transformer models pre-trained with missing features, pre-trained with imputed features and with the original features, which we assume are unavailable. Additionally, in the right panel we present standard errors for model ranks computed across all downstream tasks. From these experiments, we see that pre-training on data with imputed features significantly outperforms pre-training with missing features and that our feature imputation method enables effective transfer learning in scenarios where upstream and downstream feature sets differ in multiple features.

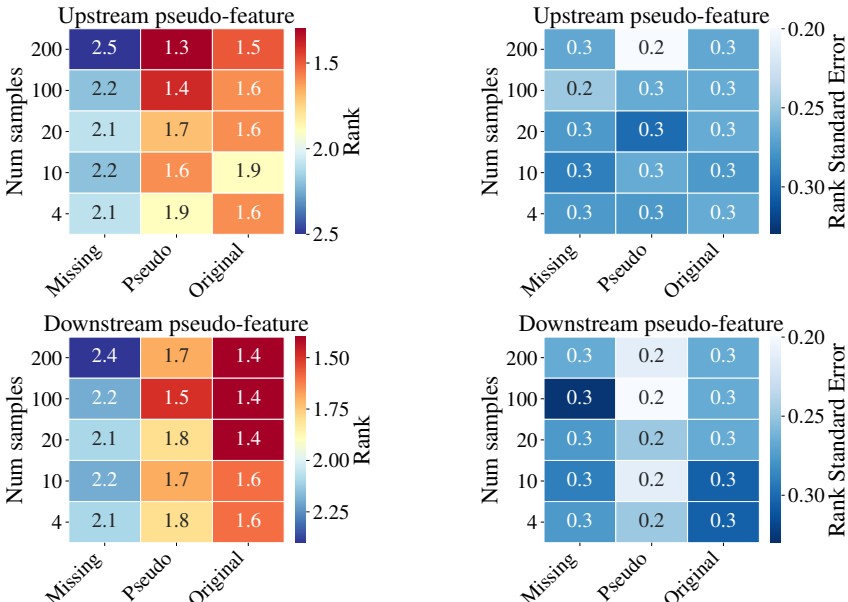

Figure 15: **Pseudo-Feature method for aligning upstream and downstream feature sets with 3 misaligned features.** Left: Comparison of ranks of FT-Transformer model trained on data with 3 missing features, with pseudo-features and with the original features. Right: Standard error for ranks computed across all tasks. Figures in the top row correspond to the experiments with missing features in the upstream data, while figures in the bottom row present experiments with missing features in the downstream data.

## D.11 ADDITIONAL EXPERIMENTS WITH CONTRASTIVE SELF-SUPERVISED LEARNING

We extend our comparison of supervised pre-training and contrastive pre-training strategies by including two additional corruption hyperparameters $\lambda = 0.7, 0.8$ for contrastive pre-training used in experiments in Section 5 (which used $\lambda = 0.9$). We also include SCARF contrastive pretraining strategy proposed in (Bahri et al., 2021). For SCARF method we generate a random mask over features, then for each masked feature we sample uniformly from the joint distribution of that feature (in the training set). Then we compute the InfoNCE loss using these corrupted inputs and the clean inputs for contrastive pretraining. We use the default corruption hyperparameter $\lambda = 0.6$ from the SCARF paper (Bahri et al., 2021). We compare all contrastive pre-training strategies and supervised pre-training in Figure 16. We find that increasing the level of corruption (i.e., decreasing $\lambda$) does not lead to improved performance on downstream tasks for contrastive pre-training. Also, while the SSL methods outperform training from scratch, especially in the lower data regimes, supervised pre-training is consistently better than all considered SSL methods.

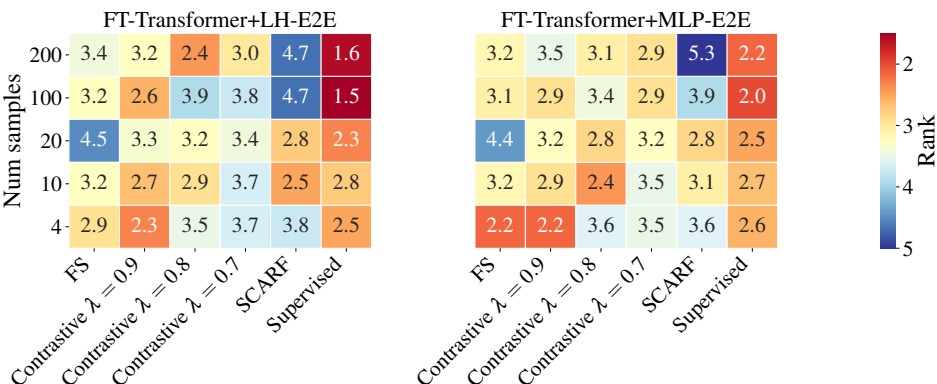

Figure 16: **Comparison of different pre-training strategies for FT-Transformer.** The left panel illustrates end-to-end fine-tuning with linear head and the right plot illustrates end-to-end fine-tuning with MLP head. Within each panel, columns represent pre-training strategies and rows correspond to the number of available downstream samples. FS stands for training from scratch, $\lambda$ denotes the corruption parameter in contrastive pre-training, SCARF corresponds to contrastive pre-training strategy from (Bahri et al., 2021) and Supervised denotes supervised pre-training on upstream data.

| Target Task | Diabetes | | | | | Hypertensive | | | | | Ischematic | | | | |
|---|---|---|---|---|---|---|---|---|---|---|---|---|---|---|---|
| Num Samples | 4 | 10 | 20 | 100 | 200 | 4 | 10 | 20 | 100 | 200 | 4 | 10 | 20 | 100 | 200 |
| FT-FS | 0.511±0.008 | 0.540±0.004 | 0.567±0.004 | 0.596±0.009 | 0.802±0.010 | 0.453±0.010 | 0.477±0.007 | 0.528±0.006 | 0.658±0.008 | 0.722±0.005 | 0.620±0.009 | 0.656±0.004 | 0.644±0.011 | 0.678±0.005 | 0.740±0.006 |
| FT-FS-2 | 0.514±0.008 | 0.550±0.006 | 0.570±0.003 | 0.666±0.006 | 0.779±0.013 | 0.464±0.006 | 0.482±0.007 | 0.525±0.004 | 0.666±0.005 | 0.701±0.004 | 0.655±0.002 | 0.656±0.005 | 0.596±0.011 | 0.740±0.003 | 0.759±0.002 |
| FT-LH-E2E | 0.661±0.007 | **0.692**±0.005 | **0.710**±0.004 | 0.770±0.006 | 0.821±0.003 | 0.669±0.005 | 0.713±0.003 | 0.707±0.002 | 0.746±0.002 | 0.760±0.002 | 0.675±0.004 | 0.664±0.005 | 0.701±0.002 | 0.771±0.002 | 0.786±0.001 |
| FT-MLP-E2EE | 0.524±0.011 | 0.606±0.009 | 0.654±0.012 | 0.781±0.007 | **0.830**±0.004 | 0.648±0.009 | 0.721±0.003 | 0.709±0.003 | **0.759**±0.003 | **0.768**±0.002 | 0.702±0.006 | 0.704±0.008 | 0.708±0.004 | 0.781±0.002 | 0.807±0.001 |
| FT-LH | **0.690**±0.005 | **0.689**±0.004 | **0.704**±0.005 | 0.700±0.005 | 0.702±0.005 | 0.658±0.008 | 0.713±0.005 | 0.708±0.004 | 0.714±0.003 | 0.716±0.003 | 0.637±0.003 | 0.679±0.002 | 0.688±0.001 | 0.716±0.002 | 0.726±0.002 |
| FT-MLP | 0.534±0.012 | 0.598±0.011 | 0.662±0.011 | 0.736±0.002 | 0.755±0.001 | 0.648±0.012 | 0.724±0.004 | 0.725±0.003 | **0.760**±0.002 | **0.767**±0.001 | 0.691±0.006 | 0.707±0.007 | 0.740±0.003 | 0.792±0.001 | 0.813±0.001 |
| ResNet-FS | 0.498±0.014 | 0.481±0.011 | 0.501±0.006 | 0.627±0.009 | 0.677±0.006 | 0.464±0.011 | 0.492±0.012 | 0.568±0.004 | 0.666±0.003 | 0.689±0.003 | 0.608±0.008 | 0.622±0.006 | 0.631±0.009 | 0.707±0.003 | 0.726±0.003 |
| ResNet-FS-2 | 0.484±0.008 | 0.514±0.005 | 0.539±0.006 | 0.677±0.003 | 0.776±0.002 | 0.500±0.007 | 0.494±0.008 | 0.538±0.006 | 0.644±0.007 | 0.724±0.002 | 0.627±0.006 | 0.646±0.005 | 0.631±0.003 | 0.736±0.002 | 0.752±0.002 |
| ResNet-LH-E2E | 0.643±0.003 | 0.660±0.003 | 0.686±0.002 | 0.705±0.002 | 0.789±0.001 | 0.684±0.004 | 0.683±0.002 | 0.697±0.001 | 0.681±0.002 | 0.762±0.001 | 0.631±0.003 | 0.663±0.002 | 0.687±0.003 | 0.767±0.001 | 0.778±0.001 |
| ResNet-MLP-E2E | 0.445±0.008 | 0.542±0.009 | 0.616±0.007 | 0.633±0.009 | 0.775±0.002 | **0.704**±0.009 | 0.719±0.003 | 0.716±0.002 | 0.737±0.003 | **0.767**±0.001 | 0.677±0.008 | 0.724±0.006 | 0.715±0.005 | 0.786±0.003 | 0.814±0.001 |
| ResNet-LH | 0.672±0.003 | 0.649±0.003 | 0.692±0.002 | 0.700±0.002 | 0.709±0.002 | 0.686±0.004 | 0.719±0.002 | 0.719±0.001 | 0.729±0.001 | 0.738±0.001 | 0.581±0.003 | 0.636±0.003 | 0.678±0.002 | 0.707±0.002 | 0.712±0.003 |
| ResNet-MLP | 0.457±0.007 | 0.546±0.009 | 0.635±0.009 | 0.720±0.002 | 0.752±0.001 | **0.704**±0.009 | 0.730±0.002 | **0.739**±0.002 | 0.757±0.002 | **0.768**±0.001 | 0.663±0.009 | 0.715±0.002 | 0.752±0.004 | **0.799**±0.001 | **0.818**±0.001 |
| MLP-FS | 0.503±0.007 | 0.491±0.005 | 0.515±0.002 | 0.652±0.003 | 0.706±0.002 | 0.442±0.003 | 0.470±0.004 | 0.578±0.001 | 0.670±0.002 | 0.699±0.002 | 0.648±0.004 | 0.643±0.008 | 0.626±0.002 | 0.728±0.002 | 0.760±0.001 |
| MLP-FS-2 | 0.507±0.005 | 0.529±0.006 | 0.543±0.004 | 0.550±0.012 | 0.671±0.005 | 0.500±0.010 | 0.492±0.011 | 0.518±0.005 | 0.638±0.004 | 0.692±0.002 | 0.639±0.004 | 0.667±0.005 | 0.668±0.003 | 0.729±0.003 | 0.761±0.003 |
| MLP-LH-E2E | 0.675±0.002 | 0.682±0.002 | 0.689±0.002 | 0.698±0.002 | 0.766±0.002 | 0.661±0.003 | 0.674±0.002 | 0.682±0.001 | 0.692±0.002 | 0.753±0.001 | 0.657±0.003 | 0.662±0.005 | 0.689±0.002 | 0.726±0.002 | 0.785±0.001 |
| MLP-MLP-E2E | 0.536±0.005 | 0.589±0.004 | 0.626±0.004 | 0.618±0.008 | 0.623±0.006 | **0.706**±0.006 | **0.736**±0.003 | 0.732±0.002 | 0.753±0.002 | 0.753±0.002 | **0.713**±0.004 | **0.745**±0.003 | **0.768**±0.002 | 0.788±0.001 | 0.811±0.000 |
| MLP-LH | 0.685±0.002 | 0.665±0.002 | 0.697±0.002 | 0.701±0.002 | 0.707±0.002 | 0.658±0.003 | 0.711±0.001 | 0.704±0.002 | 0.718±0.002 | 0.724±0.002 | 0.626±0.004 | 0.671±0.004 | 0.675±0.002 | 0.696±0.004 | 0.699±0.004 |
| MLP-MLP | 0.533±0.005 | 0.590±0.004 | 0.627±0.003 | 0.719±0.001 | 0.744±0.001 | **0.706**±0.002 | **0.738**±0.002 | **0.739**±0.002 | **0.762**±0.001 | 0.764±0.001 | **0.715**±0.005 | 0.738±0.003 | 0.758±0.002 | 0.789±0.001 | 0.807±0.001 |
| Tab-FS | 0.488±0.007 | 0.487±0.002 | 0.517±0.003 | 0.616±0.004 | 0.691±0.004 | 0.496±0.007 | 0.465±0.004 | 0.570±0.004 | 0.667±0.004 | 0.701±0.003 | 0.461±0.006 | 0.659±0.003 | 0.620±0.004 | 0.711±0.002 | 0.739±0.002 |
| Tab-FS-2 | 0.495±0.007 | 0.507±0.004 | 0.541±0.004 | 0.563±0.012 | 0.713±0.003 | 0.513±0.005 | 0.496±0.011 | 0.538±0.003 | 0.564±0.007 | 0.659±0.004 | 0.472±0.003 | 0.632±0.011 | 0.571±0.006 | 0.721±0.001 | 0.746±0.001 |
| Tab-LH-E2E | **0.693**±0.006 | **0.692**±0.006 | 0.694±0.005 | 0.704±0.006 | 0.727±0.005 | 0.658±0.004 | 0.665±0.004 | 0.685±0.003 | 0.659±0.004 | 0.727±0.003 | 0.664±0.005 | 0.668±0.005 | 0.675±0.004 | 0.708±0.004 | 0.772±0.002 |
| Tab-MLP-E2E | **0.693**±0.006 | **0.692**±0.006 | 0.694±0.005 | 0.704±0.005 | 0.725±0.005 | 0.658±0.004 | 0.665±0.004 | 0.685±0.003 | 0.659±0.004 | 0.727±0.003 | 0.664±0.005 | 0.668±0.005 | 0.675±0.004 | 0.706±0.004 | 0.770±0.002 |
| Tab-LH | **0.698**±0.006 | 0.691±0.005 | **0.703**±0.005 | 0.705±0.005 | 0.705±0.005 | 0.653±0.005 | 0.684±0.004 | 0.678±0.004 | 0.666±0.004 | 0.664±0.004 | 0.625±0.006 | 0.652±0.007 | 0.655±0.007 | 0.668±0.006 | 0.664±0.006 |
| Tab-MLP | **0.698**±0.006 | **0.698**±0.006 | **0.703**±0.006 | 0.703±0.006 | 0.702±0.006 | 0.653±0.005 | 0.658±0.004 | 0.657±0.004 | 0.656±0.004 | 0.654±0.004 | 0.625±0.006 | 0.624±0.006 | 0.628±0.006 | 0.642±0.006 | 0.639±0.006 |
| Catboost | 0.495±0.004 | 0.514±0.008 | 0.585±0.003 | 0.782±0.001 | 0.797±0.001 | 0.524±0.005 | 0.501±0.004 | 0.491±0.001 | 0.721±0.001 | 0.738±0.001 | 0.657±0.004 | 0.641±0.005 | 0.651±0.003 | 0.735±0.001 | 0.734±0.001 |
| Catboost+stacking | 0.507±0.004 | 0.555±0.006 | 0.623±0.004 | **0.807**±0.002 | **0.826**±0.001 | 0.604±0.012 | 0.602±0.008 | 0.576±0.014 | 0.732±0.001 | 0.756±0.001 | 0.671±0.004 | 0.688±0.010 | 0.734±0.004 | 0.768±0.002 | 0.766±0.002 |
| XGBoost | 0.532±0.006 | 0.525±0.003 | 0.593±0.006 | **0.805**±0.001 | 0.824±0.000 | 0.466±0.003 | 0.514±0.005 | 0.514±0.004 | 0.733±0.001 | 0.756±0.001 | 0.619±0.006 | 0.656±0.003 | 0.699±0.003 | 0.747±0.001 | 0.752±0.000 |
| XGBoost+stacking | 0.524±0.011 | 0.529±0.004 | 0.599±0.005 | **0.806**±0.000 | **0.826**±0.001 | 0.458±0.003 | 0.510±0.004 | 0.517±0.005 | 0.740±0.001 | 0.759±0.000 | 0.628±0.007 | 0.659±0.003 | 0.722±0.004 | 0.770±0.001 | 0.776±0.001 |

Table 11: **ROC-AUC scores for all models for "Diabetes", "Hypertensive" and "Ischematic" downstream tasks.** FT denotes FT-Transformer, Tab denotes TabTransformer. The first two rows in each section correspond to training from scratch, where FS corresponds to deep baseline architecture (tuned on subsample of upstream data), and FS-2 shows the results for the same architecture as one used with transfer learning. LH and MLP denote linear and MLP heads correspondingly, E2E denotes end-to-end training.

| Dataset | Heart | | | | | Overweight | | | | | Anemia | | | | |
|---|---|---|---|---|---|---|---|---|---|---|---|---|---|---|---|
| Num Samples | 4 | 10 | 20 | 100 | 200 | 4 | 10 | 20 | 100 | 200 | 4 | 10 | 20 | 100 | 200 |
| FT-FS | 0.532±0.013 | 0.557±0.009 | 0.653±0.005 | 0.687±0.008 | 0.746±0.003 | 0.516±0.012 | 0.506±0.003 | 0.623±0.006 | 0.582±0.015 | **0.822**±0.009 | 0.643±0.013 | 0.709±0.003 | 0.741±0.006 | 0.764±0.003 | 0.722±0.007 |
| FT-FS-2 | 0.517±0.005 | 0.589±0.004 | 0.571±0.011 | 0.762±0.002 | 0.748±0.003 | 0.491±0.008 | 0.490±0.009 | 0.610±0.004 | 0.638±0.005 | 0.679±0.004 | **0.728**±0.006 | 0.749±0.002 | 0.735±0.006 | 0.782±0.001 | **0.794**±0.002 |
| FT-LH-E2E | 0.589±0.002 | 0.642±0.004 | 0.667±0.002 | 0.770±0.002 | 0.782±0.002 | 0.673±0.004 | 0.620±0.008 | **0.728**±0.012 | **0.759**±0.015 | **0.818**±0.004 | 0.639±0.003 | 0.646±0.003 | 0.665±0.004 | 0.756±0.001 | 0.762±0.001 |
| FT-MLP-E2EE | 0.542±0.008 | 0.642±0.007 | 0.658±0.002 | 0.768±0.005 | 0.784±0.003 | 0.540±0.015 | 0.469±0.011 | 0.581±0.024 | 0.720±0.018 | 0.768±0.013 | 0.662±0.009 | 0.682±0.004 | 0.725±0.003 | 0.750±0.002 | 0.773±0.001 |
| FT-LH | **0.626**±0.003 | 0.664±0.002 | 0.672±0.002 | 0.689±0.003 | 0.713±0.002 | 0.669±0.002 | 0.675±0.003 | 0.682±0.002 | 0.675±0.002 | 0.674±0.002 | 0.557±0.003 | 0.580±0.003 | 0.597±0.004 | 0.610±0.003 | 0.617±0.003 |
| FT-MLP | 0.516±0.016 | 0.640±0.014 | 0.667±0.005 | **0.777**±0.001 | **0.790**±0.001 | 0.532±0.015 | 0.469±0.016 | 0.507±0.016 | 0.656±0.006 | 0.718±0.004 | 0.610±0.013 | 0.642±0.006 | 0.687±0.003 | 0.705±0.003 | 0.712±0.003 |
| ResNet-FS | 0.489±0.028 | 0.532±0.012 | 0.644±0.006 | 0.683±0.004 | 0.730±0.005 | 0.515±0.015 | 0.502±0.011 | 0.642±0.006 | 0.642±0.008 | 0.665±0.007 | 0.553±0.019 | 0.653±0.009 | 0.698±0.004 | 0.750±0.003 | 0.767±0.002 |
| ResNet-FS-2 | 0.497±0.020 | 0.551±0.010 | 0.650±0.004 | 0.750±0.004 | 0.772±0.002 | 0.495±0.009 | 0.504±0.008 | 0.615±0.004 | 0.666±0.004 | 0.739±0.003 | 0.644±0.019 | 0.752±0.004 | 0.747±0.002 | 0.757±0.002 | 0.780±0.001 |
| ResNet-LH-E2E | 0.587±0.005 | 0.609±0.004 | 0.644±0.002 | 0.656±0.002 | 0.760±0.002 | **0.699**±0.003 | **0.691**±0.003 | 0.700±0.002 | 0.714±0.001 | 0.770±0.002 | 0.607±0.003 | 0.652±0.003 | 0.665±0.002 | 0.736±0.001 | 0.748±0.001 |
| ResNet-MLP-E2E | 0.497±0.011 | 0.618±0.008 | 0.641±0.004 | 0.761±0.002 | **0.789**±0.001 | 0.547±0.008 | 0.545±0.011 | 0.568±0.013 | 0.674±0.009 | 0.765±0.005 | 0.608±0.009 | 0.697±0.008 | 0.727±0.004 | 0.760±0.001 | 0.767±0.002 |
| ResNet-LH | 0.612±0.005 | 0.657±0.004 | 0.668±0.002 | 0.661±0.003 | 0.677±0.003 | **0.698**±0.003 | 0.683±0.003 | 0.679±0.002 | 0.687±0.001 | 0.685±0.001 | 0.563±0.004 | 0.593±0.004 | 0.594±0.003 | 0.635±0.003 | 0.639±0.003 |
| ResNet-MLP | 0.488±0.012 | 0.627±0.008 | **0.683**±0.003 | 0.768±0.002 | 0.783±0.001 | 0.548±0.009 | 0.544±0.012 | 0.525±0.010 | 0.669±0.004 | 0.726±0.002 | 0.582±0.009 | 0.666±0.011 | 0.710±0.004 | 0.739±0.002 | 0.739±0.003 |
| MLP-FS | 0.426±0.007 | 0.512±0.019 | 0.646±0.002 | 0.710±0.004 | 0.765±0.001 | 0.533±0.004 | 0.499±0.008 | 0.635±0.003 | 0.651±0.005 | 0.698±0.002 | 0.641±0.006 | 0.689±0.002 | 0.708±0.000 | 0.763±0.002 | **0.796**±0.001 |
| MLP-FS-2 | 0.570±0.010 | 0.643±0.006 | 0.669±0.003 | 0.756±0.002 | 0.751±0.001 | 0.528±0.008 | 0.489±0.005 | 0.573±0.003 | 0.566±0.006 | 0.646±0.003 | **0.733**±0.005 | 0.753±0.003 | 0.738±0.001 | 0.768±0.001 | 0.790±0.001 |
| MLP-LH-E2E | **0.626**±0.003 | 0.644±0.003 | 0.665±0.002 | 0.677±0.002 | 0.766±0.001 | 0.680±0.002 | 0.675±0.002 | 0.697±0.002 | 0.683±0.002 | 0.718±0.002 | 0.610±0.003 | 0.615±0.003 | 0.657±0.002 | 0.706±0.002 | 0.749±0.000 |
| MLP-MLP-E2E | **0.629**±0.007 | **0.692**±0.003 | **0.685**±0.002 | 0.719±0.003 | 0.780±0.001 | 0.599±0.005 | 0.574±0.008 | 0.609±0.004 | 0.614±0.004 | 0.626±0.003 | 0.690±0.004 | 0.695±0.004 | 0.723±0.002 | 0.757±0.001 | 0.758±0.001 |
| MLP-LH | 0.610±0.006 | 0.650±0.002 | 0.656±0.006 | 0.665±0.004 | 0.679±0.001 | 0.675±0.002 | 0.688±0.001 | 0.685±0.001 | 0.687±0.001 | 0.686±0.002 | 0.551±0.004 | 0.567±0.004 | 0.592±0.004 | 0.597±0.003 | 0.600±0.003 |
| MLP-MLP | **0.620**±0.006 | 0.681±0.003 | **0.684**±0.002 | 0.755±0.002 | 0.767±0.001 | 0.597±0.005 | 0.576±0.005 | 0.582±0.004 | 0.646±0.003 | 0.669±0.002 | 0.655±0.005 | 0.666±0.004 | 0.694±0.004 | 0.703±0.004 | 0.699±0.002 |
| Tab-FS | 0.501±0.009 | 0.545±0.002 | 0.645±0.003 | 0.731±0.003 | 0.741±0.004 | 0.510±0.012 | 0.506±0.003 | 0.582±0.005 | 0.578±0.002 | 0.654±0.005 | 0.548±0.009 | 0.677±0.002 | 0.708±0.003 | 0.762±0.002 | 0.770±0.001 |
| Tab-FS-2 | 0.517±0.006 | 0.565±0.006 | 0.614±0.004 | 0.709±0.006 | 0.736±0.004 | 0.518±0.009 | 0.499±0.007 | 0.547±0.004 | 0.564±0.012 | 0.612±0.008 | 0.602±0.004 | 0.689±0.006 | 0.729±0.004 | 0.727±0.005 | 0.779±0.001 |
| Tab-LH-E2E | 0.602±0.006 | 0.613±0.005 | 0.636±0.004 | 0.661±0.006 | 0.760±0.004 | 0.676±0.004 | 0.679±0.004 | 0.684±0.004 | 0.684±0.003 | 0.726±0.003 | 0.594±0.004 | 0.603±0.004 | 0.650±0.004 | 0.701±0.004 | 0.742±0.003 |
| Tab-MLP-E2E | 0.602±0.006 | 0.613±0.005 | 0.636±0.004 | 0.705±0.005 | 0.757±0.004 | 0.676±0.004 | 0.679±0.004 | 0.684±0.004 | 0.684±0.003 | 0.733±0.003 | 0.594±0.005 | 0.603±0.004 | 0.650±0.004 | 0.698±0.004 | 0.745±0.003 |
| Tab-LH | 0.599±0.007 | 0.619±0.006 | 0.621±0.006 | 0.631±0.006 | 0.636±0.007 | 0.673±0.004 | 0.681±0.004 | 0.681±0.004 | 0.679±0.004 | 0.679±0.004 | 0.561±0.005 | 0.567±0.006 | 0.591±0.004 | 0.592±0.004 | 0.590±0.003 |
| Tab-MLP | 0.599±0.007 | 0.595±0.008 | 0.600±0.007 | 0.616±0.007 | 0.617±0.007 | 0.673±0.004 | 0.674±0.004 | 0.674±0.004 | 0.676±0.004 | 0.676±0.004 | 0.561±0.006 | 0.561±0.006 | 0.572±0.004 | 0.576±0.004 | 0.574±0.004 |
| Catboost | 0.554±0.005 | 0.581±0.004 | 0.652±0.001 | 0.712±0.001 | 0.744±0.001 | 0.605±0.007 | 0.499±0.008 | 0.625±0.004 | 0.764±0.002 | 0.765±0.001 | 0.560±0.009 | **0.765**±0.003 | 0.760±0.001 | 0.775±0.001 | 0.778±0.000 |
| Catboost+stacking | 0.555±0.005 | 0.624±0.009 | 0.674±0.001 | 0.756±0.001 | 0.777±0.001 | 0.614±0.006 | 0.539±0.007 | 0.672±0.005 | **0.792**±0.002 | 0.797±0.001 | 0.557±0.009 | 0.754±0.003 | 0.763±0.001 | 0.781±0.001 | 0.783±0.000 |
| XGBoost | 0.470±0.016 | 0.564±0.004 | 0.646±0.005 | 0.735±0.001 | 0.757±0.001 | 0.500±0.005 | 0.540±0.004 | **0.715**±0.010 | **0.789**±0.001 | 0.766±0.001 | 0.527±0.015 | 0.754±0.001 | **0.773**±0.002 | **0.788**±0.000 | 0.783±0.000 |
| XGBoost+stacking | 0.458±0.014 | 0.567±0.004 | 0.649±0.004 | 0.742±0.001 | 0.767±0.001 | 0.511±0.008 | 0.538±0.005 | **0.711**±0.009 | **0.789**±0.001 | 0.767±0.002 | 0.504±0.009 | 0.754±0.002 | **0.772**±0.002 | **0.788**±0.000 | 0.783±0.000 |

Table 12: **ROC-AUC scores for all models for "Heart", "Overweight" and "Anemia" downstream tasks.** FT denotes FT-Transformer, Tab denotes TabTransformer. The first two rows in each section correspond to training from scratch, where FS corresponds to deep baseline architecture (tuned on subsample of upstream data), and FS-2 shows the results for the same architecture as one used with transfer learning. LH and MLP denote linear and MLP heads correspondingly, E2E denotes end-to-end training.

| Dataset | Respiratory | | | | | Hypotension | | | | | Lipoid | | | | |
|---|---|---|---|---|---|---|---|---|---|---|---|---|---|---|---|
| Num Samples | 4 | 10 | 20 | 100 | 200 | 4 | 10 | 20 | 100 | 200 | 4 | 10 | 20 | 100 | 200 |
| FT-FS | 0.500±0.009 | 0.475±0.005 | 0.514±0.006 | 0.572±0.009 | 0.561±0.004 | 0.526±0.008 | 0.518±0.009 | 0.546±0.006 | 0.601±0.009 | 0.644±0.002 | 0.502±0.006 | 0.552±0.011 | 0.564±0.006 | 0.640±0.012 | 0.675±0.004 |
| FT-FS-2 | 0.495±0.005 | 0.469±0.005 | 0.484±0.006 | 0.562±0.010 | 0.577±0.003 | 0.574±0.003 | 0.552±0.005 | 0.539±0.012 | 0.546±0.011 | 0.641±0.001 | 0.520±0.004 | 0.530±0.002 | 0.574±0.003 | 0.598±0.003 | 0.660±0.004 |
| FT-LH-E2E | 0.545±0.004 | 0.498±0.004 | 0.517±0.002 | 0.578±0.002 | 0.585±0.001 | 0.564±0.002 | 0.564±0.003 | 0.584±0.002 | 0.592±0.002 | 0.635±0.001 | 0.584±0.006 | 0.565±0.008 | 0.630±0.004 | 0.727±0.002 | **0.745**±0.002 |
| FT-MLP-E2EE | 0.531±0.004 | 0.462±0.007 | 0.496±0.006 | 0.576±0.004 | 0.562±0.004 | 0.595±0.002 | 0.595±0.004 | 0.612±0.001 | 0.627±0.002 | 0.647±0.002 | 0.429±0.009 | 0.475±0.012 | 0.633±0.003 | 0.714±0.004 | 0.737±0.001 |
| FT-LH | 0.556±0.002 | 0.536±0.002 | 0.543±0.002 | 0.565±0.002 | 0.564±0.002 | 0.511±0.002 | 0.533±0.002 | 0.545±0.002 | 0.538±0.002 | 0.553±0.002 | **0.631**±0.004 | 0.582±0.006 | 0.655±0.003 | 0.707±0.001 | 0.724±0.001 |
| FT-MLP | 0.547±0.004 | 0.490±0.008 | 0.518±0.004 | 0.575±0.002 | 0.569±0.001 | 0.590±0.003 | 0.593±0.007 | 0.604±0.002 | 0.612±0.001 | 0.637±0.001 | 0.415±0.009 | 0.531±0.014 | 0.689±0.002 | 0.735±0.002 | 0.740±0.001 |
| ResNet-FS | 0.499±0.007 | 0.500±0.004 | 0.520±0.010 | 0.564±0.004 | 0.550±0.004 | 0.516±0.008 | 0.502±0.011 | 0.506±0.011 | 0.572±0.006 | 0.578±0.011 | 0.503±0.007 | 0.574±0.003 | 0.581±0.004 | 0.595±0.004 | 0.649±0.002 |
| ResNet-FS-2 | 0.479±0.008 | 0.460±0.005 | 0.465±0.004 | 0.565±0.003 | 0.586±0.003 | 0.552±0.010 | 0.535±0.006 | 0.539±0.004 | 0.589±0.004 | 0.638±0.002 | 0.505±0.009 | 0.551±0.007 | 0.574±0.003 | 0.523±0.018 | 0.664±0.003 |
| ResNet-LH-E2E | 0.536±0.003 | 0.530±0.002 | 0.517±0.002 | 0.577±0.001 | 0.581±0.001 | 0.521±0.005 | 0.535±0.004 | 0.531±0.003 | 0.547±0.002 | 0.590±0.001 | 0.561±0.004 | 0.571±0.005 | 0.610±0.004 | 0.635±0.002 | 0.725±0.001 |
| ResNet-MLP-E2E | 0.524±0.006 | 0.521±0.003 | 0.511±0.003 | 0.583±0.003 | 0.578±0.002 | **0.614**±0.004 | 0.603±0.004 | 0.601±0.004 | 0.624±0.003 | 0.638±0.001 | 0.442±0.006 | 0.545±0.010 | 0.657±0.005 | 0.720±0.003 | **0.744**±0.001 |
| ResNet-LH | 0.552±0.002 | **0.551**±0.001 | **0.550**±0.001 | 0.570±0.001 | 0.565±0.001 | 0.503±0.005 | 0.543±0.004 | 0.527±0.003 | 0.524±0.003 | 0.535±0.003 | 0.563±0.005 | 0.552±0.007 | 0.620±0.004 | 0.697±0.002 | 0.714±0.002 |
| ResNet-MLP | 0.520±0.006 | 0.532±0.003 | 0.530±0.003 | 0.573±0.001 | 0.566±0.001 | **0.616**±0.004 | 0.603±0.004 | 0.621±0.002 | 0.626±0.001 | 0.649±0.001 | 0.442±0.008 | 0.551±0.014 | 0.684±0.005 | **0.742**±0.001 | **0.746**±0.001 |
| MLP-FS | 0.492±0.006 | 0.494±0.004 | 0.503±0.006 | 0.564±0.003 | 0.576±0.003 | 0.504±0.004 | 0.513±0.005 | 0.529±0.005 | 0.621±0.001 | 0.648±0.002 | 0.507±0.006 | 0.531±0.015 | 0.574±0.005 | 0.598±0.004 | 0.559±0.034 |
| MLP-FS-2 | 0.500±0.003 | 0.447±0.004 | 0.458±0.003 | 0.542±0.004 | 0.574±0.004 | 0.568±0.003 | 0.545±0.005 | 0.587±0.002 | 0.571±0.007 | **0.652**±0.001 | 0.510±0.005 | 0.577±0.005 | 0.574±0.002 | 0.633±0.003 | 0.665±0.003 |
| MLP-LH-E2E | 0.545±0.002 | 0.535±0.001 | 0.531±0.001 | 0.578±0.001 | **0.591**±0.001 | 0.532±0.002 | 0.530±0.002 | 0.560±0.002 | 0.563±0.001 | 0.595±0.001 | 0.602±0.003 | **0.607**±0.003 | 0.624±0.002 | 0.624±0.003 | 0.713±0.001 |
| MLP-MLP-E2E | **0.564**±0.003 | 0.533±0.005 | 0.535±0.003 | 0.583±0.002 | 0.583±0.002 | **0.608**±0.004 | **0.628**±0.002 | **0.633**±0.001 | **0.635**±0.001 | **0.654**±0.000 | 0.430±0.009 | 0.592±0.007 | 0.684±0.004 | 0.708±0.003 | 0.741±0.001 |
| MLP-LH | 0.551±0.004 | 0.544±0.002 | **0.553**±0.001 | 0.567±0.001 | 0.567±0.001 | 0.513±0.003 | 0.550±0.004 | 0.557±0.004 | 0.548±0.002 | 0.555±0.002 | 0.608±0.004 | 0.577±0.004 | 0.644±0.003 | 0.696±0.001 | 0.707±0.003 |
| MLP-MLP | **0.571**±0.003 | **0.545**±0.004 | **0.554**±0.004 | 0.577±0.001 | 0.575±0.002 | **0.611**±0.002 | **0.629**±0.002 | **0.631**±0.001 | 0.628±0.001 | 0.645±0.001 | 0.429±0.009 | 0.594±0.007 | **0.701**±0.004 | **0.740**±0.001 | 0.743±0.001 |
| Tab-FS | 0.463±0.002 | 0.459±0.003 | 0.472±0.004 | 0.525±0.010 | 0.567±0.003 | 0.547±0.004 | 0.535±0.006 | 0.544±0.007 | 0.561±0.008 | 0.635±0.001 | 0.536±0.002 | 0.510±0.005 | 0.505±0.003 | 0.538±0.007 | 0.656±0.002 |
| Tab-FS-2 | 0.461±0.004 | 0.454±0.003 | 0.458±0.002 | 0.494±0.008 | 0.526±0.003 | 0.541±0.004 | 0.525±0.006 | 0.540±0.002 | 0.566±0.008 | 0.632±0.001 | 0.537±0.003 | 0.489±0.005 | 0.487±0.002 | 0.513±0.009 | 0.561±0.009 |
| Tab-LH-E2E | 0.531±0.004 | 0.522±0.004 | 0.504±0.003 | 0.551±0.004 | 0.582±0.002 | 0.526±0.004 | 0.523±0.004 | 0.540±0.003 | 0.544±0.003 | 0.577±0.004 | 0.615±0.005 | **0.614**±0.006 | 0.611±0.006 | 0.625±0.006 | 0.670±0.007 |
| Tab-MLP-E2E | 0.531±0.004 | 0.522±0.004 | 0.504±0.003 | 0.552±0.004 | 0.582±0.002 | 0.526±0.004 | 0.524±0.004 | 0.540±0.003 | 0.544±0.003 | 0.572±0.004 | 0.615±0.005 | **0.614**±0.006 | 0.611±0.006 | 0.625±0.006 | 0.668±0.007 |
| Tab-LH | 0.550±0.005 | **0.548**±0.003 | 0.551±0.004 | 0.562±0.004 | 0.562±0.004 | 0.511±0.004 | 0.534±0.004 | 0.537±0.004 | 0.533±0.004 | 0.537±0.004 | 0.617±0.006 | 0.595±0.006 | 0.619±0.007 | 0.647±0.008 | 0.651±0.008 |
| Tab-MLP | 0.550±0.005 | **0.547**±0.005 | **0.552**±0.004 | 0.558±0.004 | 0.558±0.004 | 0.511±0.004 | 0.512±0.004 | 0.520±0.004 | 0.523±0.004 | 0.526±0.004 | 0.617±0.006 | **0.618**±0.006 | 0.621±0.006 | 0.634±0.007 | 0.635±0.007 |
| Catboost | 0.482±0.001 | 0.492±0.003 | 0.496±0.003 | 0.554±0.002 | 0.572±0.002 | 0.523±0.001 | 0.490±0.004 | 0.531±0.005 | 0.552±0.001 | 0.635±0.001 | 0.523±0.014 | 0.536±0.002 | 0.547±0.005 | 0.618±0.006 | 0.693±0.001 |
| Catboost+stacking | 0.488±0.003 | 0.498±0.002 | 0.503±0.003 | **0.592**±0.002 | 0.584±0.001 | 0.568±0.002 | 0.501±0.004 | 0.545±0.004 | 0.583±0.001 | 0.643±0.001 | 0.515±0.017 | 0.536±0.002 | 0.555±0.007 | 0.669±0.005 | 0.718±0.002 |
| XGBoost | 0.493±0.004 | 0.493±0.003 | 0.494±0.001 | 0.571±0.003 | 0.496±0.003 | 0.507±0.004 | 0.509±0.002 | 0.553±0.002 | 0.539±0.001 | 0.517±0.009 | 0.390±0.008 | 0.483±0.004 | 0.539±0.001 | 0.558±0.002 | 0.695±0.001 |
| XGBoost+stacking | 0.499±0.005 | 0.495±0.004 | 0.497±0.002 | 0.578±0.001 | 0.499±0.003 | 0.507±0.006 | 0.511±0.003 | 0.553±0.002 | 0.543±0.001 | 0.516±0.008 | 0.394±0.006 | 0.487±0.005 | 0.545±0.002 | 0.576±0.001 | 0.723±0.001 |

Table 13: **ROC-AUC scores for all models for "Respiratory", "Hypotension" and "Lipoid" downstream tasks.** FT denotes FT-Transformer, Tab denotes TabTransformer. The first two rows in each section correspond to training from scratch, where FS corresponds to deep baseline architecture (tuned on subsample of upstream data), and FS-2 shows the results for the same architecture as one used with transfer learning. LH and MLP denote linear and MLP heads correspondingly, E2E denotes end-to-end training.

| Dataset | Atrial | | | | | Purpura | | | | | Alcohol | | | | |
|---|---|---|---|---|---|---|---|---|---|---|---|---|---|---|---|
| Num Samples | 4 | 10 | 20 | 100 | 200 | 4 | 10 | 20 | 100 | 200 | 4 | 10 | 20 | 100 | 200 |
| FT-FS | 0.603±0.008 | 0.594±0.015 | 0.617±0.007 | 0.702±0.008 | 0.690±0.008 | 0.711±0.007 | 0.703±0.010 | 0.737±0.005 | 0.870±0.007 | 0.851±0.009 | 0.593±0.008 | 0.570±0.010 | 0.600±0.005 | 0.666±0.006 | 0.640±0.005 |
| FT-FS-2 | 0.606±0.003 | 0.628±0.003 | 0.382±0.005 | 0.665±0.005 | 0.703±0.001 | 0.763±0.003 | 0.745±0.003 | 0.451±0.012 | 0.778±0.004 | 0.852±0.006 | 0.603±0.004 | 0.584±0.003 | 0.588±0.007 | 0.635±0.006 | 0.633±0.005 |
| FT-LH-E2E | 0.579±0.005 | 0.603±0.005 | 0.604±0.004 | 0.725±0.002 | 0.788±0.002 | 0.643±0.002 | 0.641±0.002 | 0.664±0.004 | 0.784±0.005 | 0.777±0.005 | 0.585±0.005 | 0.573±0.004 | 0.616±0.005 | 0.667±0.003 | 0.700±0.005 |
| FT-MLP-E2EE | 0.605±0.009 | 0.627±0.006 | 0.636±0.005 | 0.753±0.002 | 0.752±0.002 | 0.705±0.007 | 0.672±0.006 | 0.695±0.005 | 0.629±0.014 | 0.798±0.003 | 0.658±0.007 | 0.589±0.006 | 0.622±0.003 | 0.677±0.004 | 0.700±0.002 |
| FT-LH | 0.543±0.003 | 0.568±0.003 | 0.577±0.002 | 0.610±0.002 | 0.612±0.002 | 0.511±0.002 | 0.538±0.002 | 0.543±0.002 | 0.561±0.004 | 0.552±0.004 | 0.462±0.003 | 0.520±0.003 | 0.513±0.004 | 0.477±0.004 | 0.475±0.004 |
| FT-MLP | 0.587±0.015 | 0.637±0.016 | 0.667±0.007 | 0.766±0.002 | 0.772±0.001 | 0.647±0.010 | 0.563±0.014 | 0.615±0.008 | 0.715±0.004 | 0.723±0.003 | 0.685±0.005 | 0.606±0.011 | 0.666±0.006 | 0.685±0.003 | 0.703±0.003 |
| ResNet-FS | 0.595±0.007 | 0.570±0.009 | 0.599±0.006 | 0.691±0.004 | 0.625±0.019 | 0.643±0.007 | 0.708±0.007 | 0.689±0.022 | 0.755±0.007 | 0.659±0.021 | 0.560±0.015 | 0.550±0.004 | 0.611±0.009 | 0.680±0.006 | 0.700±0.007 |
| ResNet-FS-2 | 0.585±0.010 | 0.573±0.009 | 0.629±0.005 | 0.705±0.002 | 0.749±0.002 | 0.684±0.010 | 0.702±0.006 | 0.730±0.004 | 0.769±0.004 | 0.775±0.003 | 0.567±0.011 | 0.573±0.006 | 0.616±0.006 | 0.702±0.003 | 0.745±0.003 |
| ResNet-LH-E2E | 0.609±0.006 | 0.523±0.004 | 0.587±0.002 | 0.681±0.001 | 0.758±0.001 | 0.562±0.006 | 0.599±0.004 | 0.618±0.002 | 0.687±0.002 | 0.715±0.002 | 0.473±0.004 | 0.504±0.003 | 0.576±0.002 | 0.603±0.003 | 0.704±0.003 |
| ResNet-MLP-E2E | 0.671±0.013 | 0.627±0.010 | 0.650±0.005 | 0.732±0.005 | 0.748±0.005 | 0.687±0.009 | 0.652±0.011 | 0.697±0.005 | 0.729±0.005 | 0.761±0.004 | 0.715±0.007 | 0.621±0.009 | 0.630±0.005 | 0.689±0.003 | 0.720±0.002 |
| ResNet-LH | 0.582±0.007 | 0.557±0.003 | 0.570±0.002 | 0.587±0.002 | 0.595±0.002 | 0.533±0.008 | 0.560±0.005 | 0.535±0.003 | 0.542±0.004 | 0.546±0.004 | 0.436±0.005 | 0.471±0.005 | 0.488±0.005 | 0.464±0.003 | 0.463±0.003 |
| ResNet-MLP | 0.669±0.014 | 0.625±0.009 | 0.677±0.004 | 0.770±0.002 | 0.766±0.002 | 0.665±0.011 | 0.617±0.014 | 0.659±0.008 | 0.740±0.003 | 0.751±0.003 | 0.719±0.008 | 0.617±0.009 | 0.649±0.003 | 0.661±0.002 | 0.689±0.002 |
| MLP-FS | 0.535±0.015 | 0.529±0.010 | 0.603±0.005 | 0.704±0.001 | 0.707±0.001 | 0.559±0.026 | 0.570±0.022 | 0.689±0.007 | 0.754±0.002 | 0.749±0.003 | 0.518±0.013 | 0.504±0.011 | 0.584±0.009 | 0.618±0.002 | 0.632±0.003 |
| MLP-FS-2 | 0.588±0.004 | 0.594±0.008 | 0.635±0.003 | 0.577±0.010 | 0.726±0.001 | 0.764±0.003 | 0.696±0.006 | 0.733±0.003 | 0.716±0.006 | 0.750±0.002 | 0.514±0.008 | 0.505±0.006 | 0.531±0.005 | 0.626±0.005 | 0.556±0.005 |
| MLP-LH-E2E | 0.563±0.004 | 0.584±0.004 | 0.606±0.003 | 0.646±0.003 | 0.733±0.002 | 0.594±0.003 | 0.595±0.002 | 0.642±0.003 | 0.595±0.003 | 0.723±0.002 | 0.483±0.003 | 0.473±0.002 | 0.496±0.002 | 0.492±0.002 | 0.639±0.002 |
| MLP-MLP-E2E | 0.662±0.007 | 0.689±0.003 | 0.681±0.001 | 0.714±0.001 | 0.753±0.001 | 0.699±0.004 | 0.688±0.003 | 0.725±0.004 | 0.692±0.004 | 0.772±0.001 | 0.639±0.008 | 0.517±0.006 | 0.555±0.004 | 0.581±0.004 | 0.658±0.003 |
| MLP-LH | 0.535±0.005 | 0.595±0.004 | 0.588±0.004 | 0.598±0.004 | 0.593±0.004 | 0.503±0.004 | 0.561±0.004 | 0.564±0.004 | 0.565±0.004 | 0.563±0.004 | 0.460±0.004 | 0.473±0.004 | 0.459±0.004 | 0.457±0.002 | 0.454±0.002 |
| MLP-MLP | 0.653±0.003 | 0.682±0.003 | 0.679±0.002 | 0.745±0.001 | 0.740±0.001 | 0.660±0.005 | 0.647±0.003 | 0.662±0.003 | 0.728±0.002 | 0.738±0.002 | 0.642±0.006 | 0.501±0.007 | 0.523±0.006 | 0.580±0.006 | 0.590±0.006 |
| Tab-FS | 0.593±0.007 | 0.563±0.007 | 0.608±0.009 | 0.608±0.011 | 0.696±0.003 | 0.688±0.007 | 0.697±0.007 | 0.672±0.006 | 0.673±0.016 | 0.672±0.006 | 0.626±0.004 | 0.536±0.006 | 0.552±0.016 | 0.536±0.012 | 0.672±0.003 |
| Tab-FS-2 | 0.530±0.004 | 0.547±0.004 | 0.583±0.002 | 0.688±0.002 | 0.692±0.002 | 0.649±0.003 | 0.681±0.009 | 0.661±0.004 | 0.654±0.003 | 0.723±0.003 | 0.623±0.007 | 0.573±0.019 | 0.459±0.003 | 0.558±0.025 | 0.666±0.005 |
| Tab-LH-E2E | 0.561±0.008 | 0.573±0.007 | 0.591±0.005 | 0.607±0.008 | 0.716±0.005 | 0.572±0.005 | 0.579±0.005 | 0.617±0.004 | 0.599±0.006 | 0.680±0.005 | 0.506±0.007 | 0.495±0.006 | 0.476±0.005 | 0.614±0.006 | 0.666±0.006 |
| Tab-MLP-E2E | 0.561±0.008 | 0.573±0.007 | 0.591±0.005 | 0.607±0.008 | 0.720±0.005 | 0.572±0.005 | 0.579±0.005 | 0.617±0.004 | 0.599±0.006 | 0.687±0.005 | 0.506±0.007 | 0.495±0.006 | 0.476±0.005 | 0.613±0.006 | 0.668±0.006 |
| Tab-LH | 0.540±0.008 | 0.578±0.008 | 0.575±0.007 | 0.587±0.008 | 0.584±0.008 | 0.508±0.005 | 0.565±0.006 | 0.562±0.006 | 0.557±0.006 | 0.552±0.006 | 0.473±0.008 | 0.502±0.009 | 0.489±0.009 | 0.494±0.009 | 0.489±0.009 |
| Tab-MLP | 0.540±0.008 | 0.555±0.008 | 0.557±0.008 | 0.568±0.008 | 0.567±0.008 | 0.508±0.005 | 0.529±0.005 | 0.535±0.005 | 0.537±0.005 | 0.534±0.005 | 0.473±0.008 | 0.474±0.007 | 0.467±0.007 | 0.484±0.008 | 0.478±0.008 |
| Catboost | 0.598±0.001 | 0.588±0.007 | 0.541±0.004 | 0.624±0.001 | 0.705±0.002 | 0.651±0.002 | 0.618±0.005 | 0.677±0.006 | 0.834±0.003 | 0.875±0.001 | 0.541±0.002 | 0.568±0.004 | 0.611±0.004 | 0.640±0.002 | 0.706±0.001 |
| Catboost+stacking | 0.610±0.001 | 0.610±0.007 | 0.555±0.005 | 0.678±0.001 | 0.749±0.002 | 0.681±0.003 | 0.641±0.005 | 0.680±0.009 | 0.836±0.003 | 0.878±0.001 | 0.545±0.002 | 0.566±0.007 | 0.599±0.005 | 0.625±0.003 | 0.692±0.001 |
| XGBoost | 0.573±0.009 | 0.632±0.002 | 0.586±0.002 | 0.626±0.001 | 0.583±0.001 | 0.572±0.006 | 0.646±0.002 | 0.762±0.003 | 0.804±0.001 | 0.877±0.003 | 0.618±0.006 | 0.548±0.001 | 0.613±0.002 | 0.659±0.001 | 0.502±0.002 |
| XGBoost+stacking | 0.584±0.007 | 0.636±0.004 | 0.584±0.002 | 0.627±0.001 | 0.574±0.008 | 0.568±0.004 | 0.652±0.002 | 0.763±0.003 | 0.802±0.001 | 0.878±0.003 | 0.623±0.004 | 0.546±0.001 | 0.613±0.002 | 0.659±0.001 | 0.505±0.003 |

Table 14: **ROC-AUC scores for all models for "Atrial", "Purpura" and "Alcohol" downstream tasks.** FT denotes FT-Transformer, Tab denotes TabTransformer. The first two rows in each section correspond to training from scratch, where FS corresponds to deep baseline architecture (tuned on subsample of upstream data), and FS-2 shows the results for the same architecture as one used with transfer learning. LH and MLP denote linear and MLP heads corresponding, E2E denotes end-to-end training.

