# OpenReview forum: "Transfer Learning with Deep Tabular Models"
_ICLR.cc/2023/Conference — ICLR 2023 poster_

### Official Review · Reviewer_ADx7 · 2022-10-15

**Confidence:** 5
**Correctness:** 4
**Technical Novelty And Significance:** 2
**Empirical Novelty And Significance:** 2
**Recommendation:** 6

**Clarity, Quality, Novelty And Reproducibility:**

The paper is clearly written.

The quality is sound.

The novelty is ok but not significant given existing works about transfer learning on tables.

Reproducibility is good.

**Details Of Ethics Concerns:**

N.A.

**Strength And Weaknesses:**

Strengths:

1. The paper studies transfer learning on 12 medical targets.

2. The paper demonstrates the benefits of using supervised pretraining for tabular data.

3. A method for feature imputation is proposed.


Weaknesses:

1. All the experiments are performed in the medical domain. The 12 selected targets are closely related. Although the authors also validate on other datasets including Yeast functional genomics data and Emotions data, the targets in each dataset are too similar. Does the author test on other domains? For example, the authors can train on MetaMIMC and validate on Yeast functional genomics data.

2. Similarly, the authors should provide some analysis about the 12 targets of MetaMIMC. How correlated these targets are?

3. There are several works studying transfer learning for tabular data, e.g. TransTab (https://arxiv.org/abs/2205.09328). The novelty of this paper is ok but not significant. Although the paper cites the TransTab, it should not be regarded as concurrent work anymore because the paper is already on arxiv in May. The paper may need to carefully compare the differences with respect to these previous work.

**Summary Of The Paper:**

This paper studies transfer learning for tabular data. The authors take MetaMIMIC for experiments, where there are 12 medical targets. They train on 11 targets and test on the leaveout target. They find that transfer learning provides definitive advantages over gradient boosted decision trees. They further compare self-supervised learning and supervised pretraining on tabular and find that supervised pretraining is more helpful. Finally, they propose a new feature imputation method to learn tackle missing values.

**Summary Of The Review:**

1. The chosen targets may be too similar.
2. No analysis about the targets used.
3. Missing comparison with previous work. Unclear about the novelty.

---

> ### Author Response · Authors · 2022-11-17
> **Response to Reviewer ADx7**
>
> Thank you for your time and feedback! We note that in addition to this response we have made a general post  which contains several clarifications and new results, inspired by your comments. We respond to each of your comments below:
>
> 1. *Relatedness of pre-training and downstream tasks*
>
> The 12 MetaMIMIC targets are indeed all related in the sense that all targets correspond to medical conditions of varied similarity, yet the tasks in our experiments are not strongly correlated as can be seen from the correlation analysis which we provide in our response to your second point below.  Nonetheless, a degree of relatedness between tasks is fundamental to transfer learning, as noted by theoretical works on transfer learning [1,2] and also applied works [3,4].
>
> We have now run additional experiments exploring how relatedness of pre-training and downstream tasks impacts the effectiveness of transfer learning. In particular, we pre-train neural networks on a subset of tasks related to cardiovascular diseases and diabetes. We then show that transferring knowledge to more related tasks, which are also related to cardiovascular disorders, is more effective than to a set of less related tasks such as *anemia, respiratory, purpura, alcohol, hypotension*, which are found to be uncorrelated with pre-training tasks. We include a detailed discussion of these results in Appendix D.8.
>
> [1] Wang, Zirui. "Theoretical guarantees of transfer learning." arXiv preprint arXiv:1810.05986 (2018).
>
> [2] Nguyen, Cuong N., et al. "Generalization Bounds for Deep Transfer Learning Using Majority Predictor Accuracy." arXiv preprint arXiv:2209.05709 (2022).
>
> [3] Xuhong Li, Yves Grandvalet, Franck Davoine, Jingchun Cheng, Yin Cui, et al.. Transfer Learning in Computer Vision Tasks: Remember Where You Come From. Image and Vision Computing, Elsevier, 2020, 93, pp.103853.
>
> [4] Ruder, S., Peters, M.E., Swayamdipta, S. and Wolf, T., 2019, June. Transfer learning in natural language processing. In Proceedings of the 2019 conference of the North American chapter of the association for computational linguistics: Tutorials (pp. 15-18).
>
> 2. *Analysis on correlations between MetaMIMIC targets*
>
> We agree that analyzing correlations between the MetaMIMIC targets can provide insights into how much these tasks are related to each other. We incorporated your suggestion regarding correlation analysis and found that the average absolute correlation between targets is 0.08. We have now included more details on the MetaMIMIC data in Appendix B.3, such as descriptions of medical conditions used in tasks and analysis of pairwise correlations between targets.
>
> We would like to emphasize that although certain targets are indeed slightly correlated, this correlation alone cannot explain the success of transfer learning. If this were the case, then stacking would provide comparable boosts for GBDT models, which we show it does not.
>
> 3. *TransTab*
>
> Thank you for your point regarding the TransTab paper. We have now updated our draft to include a more detailed discussion delineating the differences between the paper and our own work.  Whereas TransTab focuses on learning from various data sources but on the same task, we focus on transferring between different tasks.
>
> TransTab is a novel neural network architecture for tabular data which handles variable-column tables. In particular, TransTab processes all categorical, numerical and binary features into single embeddings, which are then passed to a transformer module. The authors demonstrate that the proposed architecture can learn information from different data sources, in their case multiple tables with overlapping columns corresponding to the same task, namely clinical trial mortality prediction for the same drug.  We agree that TransTab is an insightful paper and is complementary to our own.  In fact, the authors conduct a single experiment transferring knowledge between completely different tasks with non-overlapping columns and demonstrate that pre-training does not yield benefits in this scenario.
>
> Thank you again for your thoughtful review. We added detailed discussions on our new experiments prompted by your review to Appendix B.3. and Appendix D.8. We made a significant effort to address each of your questions, including new results, and would appreciate it if you would consider raising your score in light of our response. Please let us know if you have additional questions we can address.

---

> > ### Author Response · Authors · 2022-11-18
> > **Following up with Reviewer ADx7**
> >
> > Thank you again for your thoughtful review.
> > Does our response help address your feedback? We would appreciate the opportunity to engage further if needed.

---

> > > ### Comment · Reviewer_ADx7 · 2022-11-24
> > > **Thanks for the response**
> > >
> > > The response has solved my concern about the correlations between the 12 targets. But could you please provide more evidence that transfer learning is applicable to other domains beyond the medical domain?

---

> > > > ### Author Response · Authors · 2022-11-28
> > > > **Thank you for your response**
> > > >
> > > > Thank you for your response. We are glad we addressed your other questions. Regarding “all the experiments are performed in the medical domain”, we would like to highlight that our submission contains experiments on emotions (music domain) and yeast (biology domain) datasets as well, not only on medical diagnosis.
> > > >
> > > > Our work does not enable transferring between completely unrelated domains which contain no features in common. Moreover, it is not clear if such transfer learning between unrelated domains can even possibly be effective, and existing theory supports the importance of the relationship between upstream and downstream tasks, as we explained and referenced in our previous response [1,2]. In contrast to signal processing applications of deep learning, such as audio, NLP, and vision, where domains are related despite often not appearing so because they possess common spatial structure like locality, tabular domains may possess nothing in common and contain no spatial structure. Prompted by your feedback, we now include a thorough discussion regarding the settings where our work is applicable in our updated draft, and we will include this update in our camera ready version.
> > > >
> > > > We also want to point out that single tabular domains such as medical diagnosis, or loan approval, or credit card fraud detection etc. are individually extremely impactful (perhaps more so than entire popular domains studied in the ML community), and we do not think that a transfer learning method must transfer between them (e.g. between medical diagnosis and loan approval) to be highly impactful.
> > > >
> > > > In light of our clarifications and experiments, we would appreciate it if you would consider increasing your score.
> > > >
> > > > [1] Wang, Zirui. "Theoretical guarantees of transfer learning." arXiv preprint arXiv:1810.05986 (2018).
> > > >
> > > > [2] Nguyen, Cuong N., et al. "Generalization Bounds for Deep Transfer Learning Using Majority Predictor Accuracy." arXiv preprint arXiv:2209.05709 (2022).

---

> > > > > ### Comment · Reviewer_ADx7 · 2022-11-28
> > > > > **Increasing score**
> > > > >
> > > > > Thanks for the comment. I will increase the score to 6

---

### Official Review · Reviewer_gnrQ · 2022-10-22

**Confidence:** 4
**Correctness:** 4
**Technical Novelty And Significance:** 4
**Empirical Novelty And Significance:** Not applicable
**Recommendation:** 8

**Clarity, Quality, Novelty And Reproducibility:**

The problem and solution are novel. The quality of the writing and explanations are good. The authors detail all hyperparameters and use Optuna to optimize all methods. I feel that this approach is fair, and reproducibility considerations are satisfied.

**Strength And Weaknesses:**

Strengths: The paper is well-written and easy to follow. The approach is simple but seems to work in many datasets and across different settings. Expireneltal evaluation is strong and demonstrates the potential of the approach.
Weaknesses: some related work on regularizations for tabular data is missing; for example

[1] Kadra et al. Regularization is all you Need: Simple Neural Nets can Excel on Tabular Data 2021

[2] Yang et al. Locally Sparse Neural Networks for Tabular Biomedical Data 2022

[3] James Fiedler, Simple modifications to improve tabular neural networks

To improve the presentation, it would be worthwhile to remind the reader what are MLM and FS in the caption of figure 3.
Also, please explain the procedure for downstream pseudo features in the caption of figure 4.

Do you have intuition why this procedure is beneficial? Was this done (or analyzed) in any prior work? This seems related to the procedure commonly used to impute missing values.


**Summary Of The Paper:**

The authors study the highly important problem of transfer learning with DNNs on tabular data. This avenue has huge potential in many practical applications. The problem and DNNs for tabular data are generally underexplored so this contribution could be important for the community. They propose a way to use DNNs for transfer tasks in several settings. The proposed schemes are compared to strong tree-based methods on many datasets, indicating the potential of transfer learning on tabular data.

**Summary Of The Review:**

Overall I enjoyed reading the paper. I think new solutions to challenges in tabular data are really important for the ML community. The authors did a good job of analyzing and presenting their work. The proposed solution is simple yet seems to be effective even when the number of labeled samples in the target distribution is not high. This can open the door to many follow-up works. I believe that the paper should be accepted.

---

> ### Author Response · Authors · 2022-11-17
> **Response to Reviewer gnrQ**
>
> Thank you for your supportive review! We are glad that you find our contribution to the problem of transfer learning on tabular data important and our experimental evaluation strong.  We note that in addition to this response we have made a general post which contains several clarifications and new results. We respond to each of your comments below:
>
> 1. *Additional related work*
>
> Thank you for pointing out the additional related work to us. While we already mention [1] and [3] in our related work section, we have now extended the discussion and also included [2].  These changes in our revised draft are highlighted in blue.
>
> 2. *Caption of Figure 3 and Figure 4*
>
> Thank you for your suggestion, we have now updated the captions of Figure 3 and Figure 4 to clarify the experimental setups.
>
> 3. *Do you have intuition why this procedure is beneficial? Was this done (or analyzed) in any prior work? This seems related to the procedure commonly used to impute missing values.*
>
> Indeed, our pseudo-feature method is related to imputation ideas as we incorporate an imputation procedure in our pipeline, and we reference the imputation literature [4,5,6] in Section 6 where we describe the method. In fact, misalignment of feature sets in upstream and downstream data can be considered an extreme version of missing values where all entries in a column are missing in the upstream or downstream dataset.
>
> Intuitively, our pseudo-feature method allows us to leverage for imputation and fine-tuning the useful information from the ground-truth values of the additional feature on the downstream task, while at the same time aligning the feature sets to enable pre-training on the augmented upstream data instead of discarding this useful information simply because the upstream data is missing this feature, giving us the best of both worlds.
>
> Thank you again for your supportive review. Please let us know if you have additional questions we can address.
>
> [1] A. Kadra, M. Lindauer, F. Hutter, and J. Grabocka. Regularization is all you need: Simple neural nets can excel on tabular data. arXiv preprint arXiv:2106.11189, 2021
>
> [2] J. Yang, O. Lindenbaum, and Y. Kluger. Locally sparse neural networks for tabular biomedical data. In International Conference on Machine Learning, pages 25123–25153. PMLR, 2022.
>
> [3] J. Fiedler. Simple modifications to improve tabular neural networks. arXiv preprint arXiv:2108.03214, 2021
>
> [4] S. Zhang, J. Zhang, X. Zhu, Y. Qin, and C. Zhang. Missing value imputation based on data clustering. In Transactions on computational science I, pages 128–138. Springer, 2008
>
> [5] A. M. Sefidian and N. Daneshpour. Missing value imputation using a novel grey based fuzzy c-means, mutual information based feature selection, and regression model. Expert Systems with Applications, 115:68–94, 2019.
>
> [6] J. Yoon, J. Jordon, and M. Schaar. Gain: Missing data imputation using generative adversarial nets. In International conference on machine learning, pages 5689–5698. PMLR, 2018

---

> > ### Comment · Reviewer_gnrQ · 2022-11-21
> > **After rebuttal**
> >
> > The authors have addressed all my concerns, and I keep my score at accept.

---

### Official Review · Reviewer_HXuq · 2022-10-24

**Confidence:** 3
**Correctness:** 3
**Technical Novelty And Significance:** 3
**Empirical Novelty And Significance:** 3
**Recommendation:** 8

**Clarity, Quality, Novelty And Reproducibility:**

The work is clear, and figures and writing are easy to understand. The work is of high quality, with clear contributions. The work's findings are mostly original, with some aspects being studied at least in part in prior works.

**Strength And Weaknesses:**

## Strengths

* The empirical evaluation in the paper is thorough and clear. The range of investigated methods (state of the art deep learning methods for tabular data, popular tree-based models, self-supervised methods) is very helpful in understanding trends and how these different strategies perform. Additionally, evaluation on multiple datasets of different types helps strengthen the empirical findings.

* The range of experiments and domains studied, together with distilled findings, is useful for practitioners to understand how best to apply these strategies in practice.

* The related work is fairly comprehensive as far as the ML space goes. There are other works that would be helpful to refer to, which are mentioned later in these comments.

## Neutral

* One could argue that the main methodological novelty in this paper comes from the last section, which concerns how to handle situations when the pretraining and finetuning tasks differ in what features they consider. Personally I think the strength of the empirical study and evaluation is valuable as a standalone contribution, even if this is the only 'new' methodological aspect.

## Weaknesses

### Details on empirical study
Given the work's main contribution is the empirical study, there are some more details that would be good to include for full completeness.
1.  A clearer discussion of how many different seeds/splits were considered in experiments.
2. Why were these particular low data regimes considered, as opposed to other values? What happens if we scale up to larger dataset sizes (even a single run with a large data scale would help round off the discussion)?
3. Was splitting into train/val/test on the upstream and downstream tasks on a per-patient level? What were upstream dataset sizes on average?
4. Dataset details are particularly crucial for the medical application domain -- could you provide a 1-2 sentence description of the tasks, and the label prevalences?
5. What do error bars on the ranks look like (even for a subset of the experiments)?

### Other related works
Although the related work surveys the ML field well, there are some specific relevant works in the medical ML literature that would be worth mentioning somewhere. They deal with structured data timeseries rather than prediction from a single vector, but they also investigate contrastive pretraining, masked imputation pretraining, and supervised pretraining:
* https://proceedings.mlr.press/v139/yeche21a.html (ICML 2021)
* https://dl.acm.org/doi/pdf/10.1145/3450439.3451877 (CHIL 2021).

### Other things
* In the contrastive experiments, how were the corruption hyperparameters chosen? These might significantly impact performance, so should they also be incorporated in the hyperparameter search?
* This ICLR paper from this year: https://openreview.net/forum?id=CuV_qYkmKb3 studied contrastive learning on tabular data, and performed comprehensive experiments. This might be a better augmentation strategy to compare with, rather than the one studied.
* In the pseudo feature experiments, what happens if more than one feature needs to be imputed? How does this impact performance? Do the experiments consider removing different features in each run? What does the variance in performance/rank look like?

**Summary Of The Paper:**

## Summary

* This paper investigates whether deep learning methods are effective for transfer learning on tabular data. In comparison to domains such as NLP and CV, tabular data transfer learning is not as well studied.

* The overall contributions of the paper are: (1) a thorough empirical evaluation of transfer learning methods on tabular data (considering gradient boosted decision trees, training deep models from random initialization, supervised pre-training, and self-supervised pre-training); (2) a strategy for transfer learning with tabular data when the pretraining and finetuning datasets differ in what features are measured; (3) a distilled set of recommendations for empirical work in this space.


## More details
* The paper outlines an empirical testbed for understanding the benefit of transfer learning with tabular data. The authors adopt a medical intensive care unit dataset, MIMIC-IV, for the primary evaluation. This dataset has structured tabular data for a number of patients together with 12 downstream labels of interest. The evaluation setup for supervised pretraining involves: (1) conducting pre-training on 11 of the labels; (2) fine-tuning on the 1 remaining label, with different finetuning dataset sizes, to gauge performance.

* There is also an evaluation on two non-medical datasets in the appendix.

* In this experimental evaluation, the authors find that supervised pre-training outperforms training strong gradient-boosed decision trees (GDBT) baselines  with tuned hyperparameters (Xgboost and Catboost). This is an interesting result, since it demonstrates that neural models can be effective for transfer learning on tabular data.

* The paper evaluates a variety of different neural network strategies in the experimental evaluation, including recently proposed strategies for tabular data such as FT-Transformer and TabTransformer, in addition to simpler methods such as MLPs and ResNets. A variety of different finetuning strategies are also considered for these models (e.g., linear evaluation, end2end training with a linear head/MLP head).

*  The second part of the experimental evaluation compares different pre-training strategies, namely supervised pre-training as compared to self-supervised strategies such as masked imputation. Overall, supervised pretraining strategies perform better than self-supervised pretraining methods.

* The paper then studies a setting where the pretraining and finetuning sets may contain different features, so straightforward transfer learning is not as readily appropriate. To tackle this, the paper suggests a process to impute the missing feature using an additional trained model, before re-running pretraining/finetuning. They find that this strategy improves over just excluding the missing feature (though the evaluation in this setting appears less complete).

* The paper presents detailed information about experimental setups, including considerations of: how to handle the small data regimes, how to do hyperparameter search, ranges of considered hyperparameters, how early stopping applies in such extreme small data regimes.

**Summary Of The Review:**

Overall, I think the experimental evaluation in this paper is thorough and has useful insights for practitioners. I would therefore vote to accept it.

---

> ### Author Response · Authors · 2022-11-17
> **Response to Reviewer HXuq Part 1**
>
> Thank you for your thoughtful review and encouraging feedback! We appreciate that you find our results interesting and our empirical evaluation thorough and clear. We note that in addition to this response we have made a general post which contains several clarifications and new results. We respond to each of your comments below:
>
> 1. *Discussion of how many different seeds/splits were considered in experiments.*
>
> Thank you for pointing this out. We have now updated our draft to include more details on the number of splits and seeds in Appendix B.4. The new content is highlighted in blue. To summarize, we run each experiment with 10 random seeds for the experiments in Section 4 and 5 random seeds for the experiments in Sections 5, 6. The train/val/test split does not change across the seeds.
>
> 2. *What happens if we scale up to larger dataset sizes?*
>
> We initially chose small downstream dataset sizes since this is the regime where neural networks typically struggle in the tabular domain, but we agree with you that larger dataset sizes are important to test too.  Prompted by your question, we now ran additional experiments with higher downstream data availability. In particular, we consider scenarios where we have 400,1000, and 2000 downstream samples. Interestingly, we find that even in higher data regimes, FT-Transformer with transfer learning outperforms GBDT with stacking, while MLPs perform on par with CatBoost. We include these results as well as experimental details in Appendix D.9.
>
> 3. *Was splitting into train/val/test on the upstream and downstream tasks on a per-patient level? What were upstream dataset sizes on average?*
>
> Yes, we split both upstream and downstream data into train/val/test on a per-patient level. Also, downstream test patients are disjoint from upstream patients so that there is no “information leakage” for models using upstream data.  The upstream dataset size is 22701 patients. We include these details in our discussion on the number of seeds and splits in Appendix B.4.
>
> 4. *Dataset details are particularly crucial for the medical application domain*
>
> Thank you for pointing this out. We agree that providing a description of medical conditions used as tasks in our experiments is important, especially in light of our additional experiments on the effectiveness of knowledge transfer to related/unrelated tasks. We include this discussion in Appendix B.3.
>
> 5. *What do error bars on the ranks look like?*
>
> We have now included a figure with standard errors computed across all downstream tasks for ranks for the main experiment on transfer learning with deep tabular models, please see Figure 7 in Appendix D.2, which complements Figure 2 in the main body.
>
> 6. *Other related works*
>
> Thank you for pointing out these papers to us. We have updated our related work section to include these works.
>
> 7. *In the contrastive experiments, how were the corruption hyperparameters chosen?*
>
> We note that since MLM pre-training requires training additional classification heads for every feature in the data, it dramatically increases the memory requirements. Because of that, we limit the experiments with self-supervised pre-training, including contrastive pre-training, to using the default FT-transformer configuration.
>
> We originally selected the corruption hyperparameters in contrastive pre-training in small-scale experiments, however, we agree that the choice of the corruption hyperparameters might significantly impact performance, so we have now conducted experiments with two additional sets of hyperparameters $\lambda = 0.7$, $\lambda = 0.8$ to verify whether increasing the level of corruption (i.e., decreasing $\lambda$) would lead to improved performance on downstream tasks for contrastive pre-training. We find that these configurations do not result in better downstream performance than our original contrastive configuration with $\lambda=0.9$ and that supervised pre-training is consistently better. We include this comparison in Appendix D.11.

---

> ### Author Response · Authors · 2022-11-17
> **Response to Reviewer HXuq Part 2**
>
> 8. *SCARF Contrastive Learning*
>
> Thank you for pointing out this work to us. We have now run additional experiments comparing the contrastive pre-training strategy from [1], which we used in our submission, SCARF, and supervised pre-training. In fact, we find that [1] results in better downstream performance compared to SCARF. One reason for that might be that the SCARF paper uses more corruption, and we observe from the experiments above that increasing corruption in contrastive learning does not lead to better performance on MetaMIMIC. We provide a detailed discussion on the comparison of SCARF, contrastive pre-training and supervised pre-training in Appendix D.11. To summarize, we find that while the SSL methods outperform training from scratch, especially in the lower data regimes, supervised pre-training is consistently better than all considered SSL methods.
>
> 9. *In the pseudo feature experiments, what happens if more than one feature needs to be imputed? How does this impact performance? Do the experiments consider removing different features in each run? What does the variance in performance/rank look like?*
>
> Prompted by your question, we have now run additional experiments for our feature imputation method, where we remove three features instead of a single feature. In our experiments, we remove features which are found to be important for a GBDT model (Catboost) to ensure that we remove useful information from the data. These important features are different across downstream tasks. From our new experiments, we find that pre-training on data with pseudo features significantly outperforms pre-training with missing features, which suggests that our pseudo feature method is useful in scenarios where upstream and downstream feature sets differ in multiple features. We include the detailed discussion and results of this experiment in Appendix D.10.
>
> Thank you again for your supportive review. We added detailed discussions on our new experiments prompted by your review to Appendix D. Please let us know if you have additional questions we can address.
>
> [1] Somepalli, Gowthami, et al. "Saint: Improved neural networks for tabular data via row attention and contrastive pre-training." arXiv preprint arXiv:2106.01342 (2021).

---

> > ### Comment · Reviewer_HXuq · 2022-11-25
> > **Response to author comments**
> >
> > Thank you very much for your detailed response and the additional clarifications. These details address the questions I had, so I will keep my score.

---

### Official Review · Reviewer_1RK9 · 2022-10-28

**Confidence:** 3
**Clarity, Quality, Novelty And Reproducibility:** Good clarity, quality and reproducibi…
**Correctness:** 3
**Technical Novelty And Significance:** 2
**Empirical Novelty And Significance:** 2
**Recommendation:** 6

**Strength And Weaknesses:**

Strength
1. Deep learning with tabular data is an important and unsolved problem.
2. This paper is clearly written.
3. The proposed approach of pseudo-value completion seems interesting and effective.

Weaknesses
1. Previous studies (such as FT-Transformer) have already shown that DNNs can generally outperform XGBoost/CatBoost. Therefore, the main findings of this paper sound less surprising. Authors provide no insights about when DNNs transfer better than tree models and when not.
2. All experiments use 11 splits as the upstream task and 1 as the downstream task. This may not be representative enough for various applications. I wonder whether using less splits in pre-training still results in the same conclusion.
3. A minor typo: “with without”

**Summary Of The Paper:**

This paper studies transfer learning on deep tabular models. Authors create a transfer learning benchmark from the MetaMIMIC repository, and then conduct transfer learning experiments on it. Specifically, authors find that fine-tuning pre-trained FT-Transformer consistently outperforms classical models such as XGBoost and Catboost with different training data sizes. Authors also find self-supervised pre-training is inferior to supervised pre-training on tabular data. Additionally, to deal with applications with missed features, authors propose to use pseudo values for feature alignments.

**Summary Of The Review:**

Overall, this paper presents useful comparisons of transfer learning between deep Transformers and tree models on tabular data, though the findings are not so surprising. My major concerns are, 1) when (on which kinds of data) shall we use deep Transformers and when to use tree models, and 2) whether is the conclusion still right with fewer pre-training tasks?

---

> ### Author Response · Authors · 2022-11-17
> **Response to Reviewer 1RK9**
>
> Thank you for your time and feedback! We note that in addition to this response, we have made a general post which contains several clarifications and new results.  Inspired by your feedback, we have run several new experiments and updated our draft accordingly.  We respond to each of your comments below:
>
> 1. *Given that FT-Transformer and NNs can outperform decision trees methods, why are our results surprising?*
>
> Recent works have actually suggested that decision tree methods such as CatBoost outperform neural networks on small, and typically even medium, sized datasets, whereas neural networks often perform better on large datasets and in some cases medium sized datasets [1,2]. In contrast, we find that transfer learning can enable neural networks to definitively outperform CatBoost and XGBoost even on very small downstream datasets, and even when decision tree methods employ stacking.  Additionally, the value of representation learning and knowledge transfer on tabular data has been widely questioned [3], and a recent review paper even points out that transfer learning for tabular data is an open problem [2].  Aside from the central contribution of showing that transfer learning gives NNs a clear edge over decision tree methods, we also develop a framework for solving a problem which is commonplace in the tabular domain, namely feature sets may vary across tasks, and we highlight strong performance in this setting. We feel that this work provides actionable insights to practitioners who are considering tabular deep learning, as pointed out by another reviewer, and we also think our work contributes to the ongoing debate concerning the value of deep learning in the tabular domain [1,2,3].
>
> [1] Grinsztajn, Léo, Edouard Oyallon, and Gaël Varoquaux. "Why do tree-based models still outperform deep learning on tabular data?." arXiv preprint arXiv:2207.08815 (2022).
>
> [2] Borisov, Vadim, et al. "Deep neural networks and tabular data: A survey." arXiv preprint arXiv:2110.01889 (2021).
>
> [3] Shwartz-Ziv, Ravid, and Amitai Armon. "Tabular data: Deep learning is not all you need." Information Fusion 81 (2022): 84-90.
>
> 2. *When do DNNs with transfer learning work better than tree models?*
>
> We hypothesize that knowledge transfer between tasks is most effective in scenarios where upstream and downstream tasks are related. Inspired by your comment, we have now run additional experiments exploring how relatedness of pre-training and downstream tasks impacts the effectiveness of transfer learning. In particular, we pre-train neural networks on a subset of tasks related to cardiovascular diseases and diabetes. We then show that transferring knowledge to more “related” tasks, which are also related to cardiovascular disorders, is more effective than to a set of less related tasks such as *anemia, respiratory, purpura, alcohol, and hypotension*, which are found to be uncorrelated with pre-training tasks. We include these results in Appendix D.8.
>
> 3. *Effectiveness of transfer learning when using fewer pre-training tasks.*
>
> Prompted by your comment, we conduct additional experiments with fewer pre-training tasks and include these results in Appendix D.7. We consider the first 5 MetaMIMIC targets: we again pre-train on all but one target and transfer knowledge to the remaining task. Even with fewer pre-training tasks, similarly to our previous results in Figure 2, we find that deep learning models with transfer learning outperform CatBoost with stacking. That is, transfer learning with deep tabular models is effective, more so than strong baselines, even with fewer pre-training tasks (provided that pre-training tasks are related to the downstream tasks as shown by our experiment above).
>
> 4. *Typo* Thank you for pointing out the typo, we corrected it in our revised draft.
>
>
> Thank you again for your thoughtful review. We added detailed discussions on our new experiments prompted by your review to Appendix D. We made a significant effort to address each of your questions and would appreciate it if you would consider raising your score in light of our response. Please let us know if you have additional questions we can address.

---

> > ### Author Response · Authors · 2022-11-18
> > **Following up with Reviewer 1RK9**
> >
> > Thank you again for your thoughtful review.
> > Does our response help address your feedback? We would appreciate the opportunity to engage further if needed.

---

> > ### Comment · Reviewer_1RK9 · 2022-11-24
> > **Thank you for the response**
> >
> > I still think the finding that "DNNs outperform decision tree methods in transfer learning" is less significant. Though some existing works discussed the comparison between them, these studies are not targeted on transfer learning. It is well known that DNNs can be fine-tuned, but tree models are just frozen feature extractors (learned on upstream tasks) regarding transfer learning. So, it's not so surprising that DNNs perform better in transfer learning. Moreover, transferring DNNs cannot consistently outperform transferring tree models, especially when the upstream and downstream tasks are less relevant, as shown in Figure 13. Thereby, we can at most regard transferring DNNs as "generally superior", but the model choice still highly depends on specific data. In practice, we may still need to try different models in most cases.

---

> > > ### Author Response · Authors · 2022-11-28
> > > **Thank you for your response**
> > >
> > > We thank you for your response and willingness to engage, and we are glad we addressed your other questions. We want to stress that the debate between tree-based and deep learning methods for tabular data is highly topical and has resulted in impactful publications [1,2,3,4,5] and uproar on social media. This debate is not only over whether neural networks are better than CatBoost but also whether tabular neural networks have a place in the toolbox of data scientists at all and if representation learning is at all useful for tabular data [1,2,4].
> > >
> > > While the current debate has mostly only considered from-scratch training, a recent review paper specifically called for more research in transfer learning for tabular data [6], and our work shows that settings where transfer learning is possible yield a sizable advantage to NNs, even though tree-based methods are commonly used for transfer learning in real-world deployments via stacking. Additionally, the success of neural networks on tabular data has so far been achieved on larger datasets [4,5] while on smaller datasets GBDT have been shown to be dominant [5]. We show that transfer learning enables neural networks to outperform GBDT (and GBDT+stacking) on extremely small downstream datasets thus altering the current state of the GBDT vs tabular deep learning debate.
> > >
> > > While we agree that fine-tuning neural networks enables effective knowledge transfer and this ability can explain the performance boosts in hindsight, we believe that the effectiveness of transfer learning is not obvious on tabular data and needs to be investigated just as the advantage of neural networks and representation learning over GBDT is not obvious overall which is confirmed by the significant part of the community frequently challenging neural networks for tabular data.
> > >
> > > Therefore, our work both contributes to the ongoing debate surrounding deep learning for tabular data and presents a how-to guide for practitioners using stacking who could likely squeeze more juice out of their data. Prompted by your feedback, we have added a detailed discussion of how our work fits into the current conversation surrounding tabular deep learning as well as its implications for practitioners and when they should consider tabular transfer learning. We will include this update in our camera ready version as well.
> > >
> > > We appreciate this conversation, and we feel that this exchange clarifies the significance of our work to the community. We would appreciate it if you would consider increasing your score in light of our response.
> > >
> > > [1] Grinsztajn, Léo, Edouard Oyallon, and Gaël Varoquaux. "Why do tree-based models still outperform deep learning on tabular data?." arXiv preprint arXiv:2207.08815 (2022).
> > >
> > > [2] Shwartz-Ziv, Ravid, and Amitai Armon. "Tabular data: Deep learning is not all you need." Information Fusion 81 (2022): 84-90.
> > >
> > > [3] Huang, X., Khetan, A., Cvitkovic, M. and Karnin, Z., 2020. Tabtransformer: Tabular data modeling using contextual embeddings. arXiv preprint arXiv:2012.06678.
> > >
> > > [4] Gorishniy, Y., Rubachev, I., Khrulkov, V. and Babenko, A., 2021. Revisiting deep learning models for tabular data. Advances in Neural Information Processing Systems, 34, pp.18932-18943.
> > >
> > > [5] Gorishniy, Y., Rubachev, I. and Babenko, A., 2022. On Embeddings for Numerical Features in Tabular Deep Learning. arXiv preprint arXiv:2203.05556.
> > >
> > > [6] Borisov, Vadim, et al. "Deep neural networks and tabular data: A survey." arXiv preprint arXiv:2110.01889 (2021).

---

> > > > ### Comment · Reviewer_1RK9 · 2022-11-28
> > > > **Increasing my score**
> > > >
> > > > To my opinion, the significance of the debate between different methods lie in motivating deeper exploitation on those methods and specific guides in applications. A conclusion like "which one is generally better" seems less useful in practice. Moreover, considering the hierarchical and over-parameterized structure of DNNs, it sounds not surprising that DNNs are more suitable for transfer learning.
> > > >
> > > > Nevertheless, I agree that a rigorous and comprehensive empirical study on deep transfer learning with tabular data is necessary for the community. As authors proposed an effective pseudo-value based approach, and my other concerns have been addressed, I raise my score to 6.

---

### Author Response · Authors · 2022-11-17
**General Response to Reviewers and AC Part 1**

We thank the reviewers for their thoughtful and generally supportive feedback.  We are particularly encouraged that they feel our contributions are impactful, practically useful, thoroughly evaluated, and “open the door to many follow-up works.”

We also want to emphasize the timeliness of this work.  There is an ongoing debate in the community over whether deep learning is useful for tabular data, which is the dominant format for data in real-world ML [1,2,3].  Recent works suggest that neural networks are only useful in the large dataset regime [1], and others question if representation learning is really necessary for tabular data [3].  In contrast, our work demonstrates that deep learning decisively outperforms competitors such as XGBoost and CatBoost when relevant pre-training data is available, even when downstream datasets are very small.  Moreover, our work highlights the value of representation learning and knowledge transfer in the tabular domain.

On top of our contribution to the debate surrounding the merits of deep learning for tabular data, our work serves as a how-to guide for practitioners who are considering deep learning in their own tabular use case, as pointed out by reviewers. To enable transfer learning in practical settings where feature sets often differ across datasets and learning problems, we develop a new method which leverages feature imputation and showcases strong performance compared to baselines.

Inspired by feedback from the reviewers, we have now run numerous additional experiments and made corresponding edits to our updated draft as well. We detail several of these new experiments below. We hope that reviewers can consider our response in their final assessment.

1. **Analysis on when transfer learning with tabular DNNs is most effective.**
We hypothesize that knowledge transfer between tasks is most effective in scenarios where upstream and downstream tasks are related. To validate this hypothesis we run additional experiments exploring how relatedness of pre-training and downstream tasks impacts the effectiveness of transfer learning. In particular, we pre-train neural networks on a subset of tasks related to cardiovascular diseases and diabetes. We then show that transferring knowledge to more “related” tasks, which are also related to cardiovascular disorders, is more effective than to a set of less related tasks, which are found to be uncorrelated with pre-training tasks. We include these results in Appendix D.8.

2. **Effectiveness of transfer learning when using fewer pre-training tasks.**
To verify that transfer learning with DNNs works in scenarios with less pre-training signal available, we conduct additional experiments with fewer pre-training tasks and include these results in Appendix D.7. We experiment with the first 5 MetaMIMIC tasks: we again pre-train on all but one target and transfer knowledge to the remaining target. Even with fewer pre-training tasks, similarly to our previous results in Figure 2, we find that deep learning models with transfer learning outperform CatBoost with stacking. That is, transfer learning with deep tabular models is effective, more so than strong baselines, even with fewer pre-training tasks (provided that pre-training tasks are related to the downstream tasks as shown by our experiment above).

3. **Transfer Learning in higher data regimes.**
We run additional experiments with higher downstream data availability. In particular, we consider scenarios where we have 400,1000, and 2000 downstream samples. Interestingly, we find that even in higher data regimes, FT-Transformer with transfer learning outperforms GBDT with stacking, while MLPs perform on par with CatBoost. We include these results as well as experimental details in Appendix D.9.
How does the pseudo-feature method work if more than one feature needs to be imputed?
We run additional experiments for our feature imputation method, where we remove three important features instead of a single feature. From our new experiments we find that pre-training on data with imputed features significantly outperforms pre-training with missing features, which suggests that our feature imputation method works in scenarios where upstream and downstream feature sets misalign in multiple features. We include results of this experiment in Appendix D.10.

---

> ### Author Response · Authors · 2022-11-17
> **General Response to Reviewers and AC Part 2**
>
> 4. **How does the pseudo-feature method work if more than one feature needs to be imputed?**
> We run additional experiments for our feature imputation method, where we remove three important features instead of a single feature. From our new experiments we find that pre-training on data with imputed features significantly outperforms pre-training with missing features, which suggests that our feature imputation method works in scenarios where upstream and downstream feature sets misalign in multiple features. We include results of this experiment in Appendix D.10.
>
>
> 5. **SCARF Contrastive Learning.**
> We have run additional experiments comparing the contrastive pre-training strategy from [4], which we used in our submission, SCARF [5], and supervised pre-training. We find that [4] results in better downstream performance compared to SCARF. One reason for that might be that the SCARF paper uses more corruption [5], and we observe from our experiments with contrastive learning that increasing corruption in contrastive learning does not lead to better performance on MetaMIMIC. We provide a detailed discussion on the comparison of SCARF [5], contrastive pre-training [4], and supervised pre-training in Appendix D.11. To summarize, we find that while the SSL methods outperform training from scratch, especially in the lower data regimes, supervised pre-training is consistently better than all considered SSL methods.
>
>
> [1] Grinsztajn, Léo, Edouard Oyallon, and Gaël Varoquaux. "Why do tree-based models still outperform deep learning on tabular data?." arXiv preprint arXiv:2207.08815 (2022).
>
> [2] Borisov, Vadim, et al. "Deep neural networks and tabular data: A survey." arXiv preprint arXiv:2110.01889 (2021).
>
> [3] Shwartz-Ziv, Ravid, and Amitai Armon. "Tabular data: Deep learning is not all you need." Information Fusion 81 (2022): 84-90.
>
> [4] Somepalli, Gowthami, et al. "Saint: Improved neural networks for tabular data via row attention and contrastive pre-training." arXiv preprint arXiv:2106.01342 (2021).
>
> [5] Bahri, Dara, et al. "Scarf: Self-supervised contrastive learning using random feature corruption." arXiv preprint arXiv:2106.15147 (2021)

---

### Decision · Program_Chairs · 2023-01-20

**Decision:**

Accept: poster

**Justification For Why Not Higher Score:**

The limited set of domains might reduce the general interest of the paper somewhat.

**Justification For Why Not Lower Score:**

Investigating the role of pre-training adds timely new evidence to an important debate.

**Metareview: Summary, Strengths And Weaknesses:**

This paper is an empirical study of whether recently introduced methods for tabular neural networks can work better than the more established methods methods based on ensembles of trees, once we take pre-training into account.

Strengths:
* This paper adds interesting new evidence to an ongoing recent debate [1-4].
* The reviewers agreed that the empirical evaluation is thorough and clear.
* The method of "pseudo-features" shows that pre-training can be useful for data imputation as well, which is an interesting result.

Weaknesses:
* Some of the reviewers raise questions about whether the main findings of the paper. However, there is active debate in this area, so new evidence seems worthwhile.
* Reviewers also pointed out that these studies were limited to the medical domain. I agree that experiments in other domains would provide additional information that would be very useful, but there also seems to be broad agreement that the current result already make a worthwhile contribution.


[1] Grinsztajn, Léo, Edouard Oyallon, and Gaël Varoquaux. "Why do tree-based models still outperform deep learning on tabular data?." arXiv preprint arXiv:2207.08815 (2022).

[2] Borisov, Vadim, et al. "Deep neural networks and tabular data: A survey." arXiv preprint arXiv:2110.01889 (2021).

[3] Shwartz-Ziv, Ravid, and Amitai Armon. "Tabular data: Deep learning is not all you need." Information Fusion 81 (2022): 84-90.

[4] Somepalli, Gowthami, et al. "Saint: Improved neural networks for tabular data via row attention and contrastive pre-training." arXiv preprint arXiv:2106.01342 (2021).


**Note From Pc:**

if the above contains the word "oral" or "spotlight" please see: "oral" presentation means -> notable-top-5% and "spotlight" means -> notable-top-25%. As stated in our emails, we are disassociating presentation type from AC recommendations